# Long-range phase coherence and tunable second order φ₀-Josephson effect in a Dirac semimetal 1T-PtTe₂
Pranava K. Sivakumar [1] ✉, Mostafa T. Ahari[2], Jae-Keun Kim [1], Yufeng Wu[1], Anvesh Dixit[1], George J. de Coster[3], Avanindra K. Pandeya[1], Matthew J. Gilbert [2,4] & Stuart S. P. Parkin [1] ✉

Superconducting diode effects have recently attracted much attention for their potential applications in superconducting logic circuits. Several pathways have been proposed to give rise to non-reciprocal critical currents in various superconductors and Josephson junctions. In this work, we establish the presence of a large Josephson diode effect in a type-II Dirac semimetal 1T-PtTe₂ facilitated by its helical spin-momentum locking and distinguish it from extrinsic geometric effects. The magnitude of the Josephson diode effect is shown to be directly correlated to the large second-harmonic component of the supercurrent. We denote such junctions, where the relative phase between the two harmonics can be tuned by a magnetic field, as 'tunable second order φ₀-junctions'. The direct correspondence between the second harmonic supercurrents and the diode effect in 1T-PtTe₂ junctions at relatively low magnetic fields makes it an ideal platform to study the Josephson diode effect and Cooper quartet transport in Josephson junctions.

Non-reciprocal electrical transport in materials with broken inversion symmetry manifests itself as non-linear responses in electrical conductivity when time-reversal symmetry is also broken[1]. These responses are usually quantified through what is known as the magnetochiral anisotropy (MCA)[2–4]. It was observed that the MCA was strongly enhanced in the superconducting state of non-centrosymmetric systems, promoting the search for non-reciprocal effects in superconducting systems[5–7]. Non-reciprocal critical currents titled the superconducting diode effect was first observed along with large MCA in a non-centrosymmetric thin film superlattice of superconductors[8]. Since its discovery, over the past few years, there has been a lot of interest in creating and understanding superconducting and Josephson diodes composed of various materials, from both a fundamental and technological perspective[8–17]. These devices exhibit non-reciprocal superconducting critical currents and allow for unidirectional propagation of supercurrents and normal currents in the opposite direction, which is quite promising for the creation of various low dissipative technologies. The observation of a supercurrent diode effect requires the breaking of both inversion and time-reversal symmetries (TRS)[18], which also makes it a useful 'tool' providing insights into a material's broken symmetries and other properties in the superconducting state such as the nature of spin-orbit coupling[14] and in the determination of a chiral superconducting state that breaks time-reversal symmetry[19]. There have been multiple pathways proposed for the creation of the supercurrent diode effect, where most of them rely on the creation of Cooper pairs with non-zero momentum either due to the presence of spin-momentum locking in the material or Meissner screening currents induced by the magnetic field[13,17,20–24]. However, experimentally disentangling these two effects has been challenging.

In this paper, we perform a detailed study of the Josephson diode effect (JDE or $\Delta I_c$) in a transition metal dichalcogenide and Dirac semimetal system (1T-PtTe₂) in different current and magnetic field geometries. This allows for distinguishing between intrinsic contributions to the JDE arising from the band structure and extrinsic junction geometric effects and establish the presence of helical spin-momentum locking in the system. The supercurrent behavior in the junction is studied in detail by considering a current-phase relationship (CPR) with a second harmonic term that we refer to as a 'tunable second order φ₀-junction' CPR. The observations from this CPR are verified by measuring the evolution of critical currents in PtTe₂ junctions in the presence of a magnetic flux and a magnetic field that is needed to induce the JDE. These measurements are used to provide direct evidence that the oscillations in $\Delta I_c$ are second harmonic in nature with nodes occurring at every half-magnetic flux quantum $\left(\frac{\varphi_0}{2}\right)$ and that the magnitude of $\Delta I_c$ is closely related to the magnitude of second harmonic supercurrents in the system and a phase difference ($\delta$) between the first and

[1]Max Planck Institute of Microstructure Physics, 06120 Halle (Saale), Germany. [2]Materials Research Laboratory, The Grainger College of Engineering, University of Illinois, Urbana-Champaign, Illinois, 61801, USA. [3]DEVCOM Army Research Laboratory, 2800 Powder Mill Rd, Adelphi, MD, 20783, USA. [4]Department of Electrical Engineering, University of Illinois, Urbana-Champaign, IL, 61801, USA. ✉e-mail: sivakumar@mpi-halle.mpg.de; stuart.parkin@mpi-halle.mpg.de

second harmonic components, as predicted from the CPR. This CPR combined with the tunability of $\delta$ with a magnetic field provides the possibility of controlling the relative magnitudes and direction of first- and second-harmonic supercurrents leading to controllable flow of Cooper pairs and Cooper quartets. Based on the transparency of the junctions studied, we also comment on the potential contribution of Meissner screening currents to the observed JDE, as compared to the helical spin-momentum locking in PtTe$_2$. Finally, the role of the helical spin-momentum locked topological states in the formation of high transparency interfaces and phase coherent higher order Andreev reflections in PtTe$_2$ junctions that leads to the presence of a strong second harmonic term and hence a large JDE in the system is discussed.

## Results

### Lateral Josephson junctions of PtTe$_2$

1T-PtTe$_2$ is an air stable two-dimensional Van der Waals transition metal dichalcogenide (TMDC) that crystallizes in the centrosymmetric $P\bar{3}m1$ crystal structure [Fig. 1a]. Though 1T structures in which the transition metal atom has an octahedral coordination are centrosymmetric down to the monolayer limit, they have local inversion symmetry breaking within a single layer at the chalcogenide sites, giving the transition metal atom a $D_{3d}$ point group symmetry and the chalcogenide atom a $C_{3v}$ point group symmetry. This local inversion symmetry breaking gives rise to a series of band inversions and topological surface states[25] in these materials along with

Rashba spin-orbit coupling of equal magnitude on each of the chalcogenide layers within each monolayer, with the top and bottom chalcogenide atomic layers having opposite spin-orbit coupling strengths. These differences in point group symmetry are predicted to give rise to a layer dependent 'local Rashba effect' with helical spin-momentum locking of opposite helicities on alternating chalcogenide layers as has been observed through spin-polarization measured through spin- and angle-resolved photoemission measurements in certain group-X transition metal dichalcogenides including PtTe$_2$[26], but also, for example, PtSe$_2$[27] and PdTe$_2$[25] as well as in the cuprate superconductor Bi$_2$Sr$_2$CaCu$_2$O$_{8+x}$[28]. This helical spin-momentum locking in 1T structures is analogous to Ising spin-momentum locking in 2H TMDCs[29,30]. In the case of PtTe$_2$, Dirac cone-like dispersions with helical spin texture and net spin polarization have been observed near the Fermi level through angle- and spin-resolved photoemission[26].

Thin PtTe$_2$ flakes are exfoliated from a single crystal on a Si/SiO$_2$ substrate and lateral Josephson junctions are fabricated by depositing Ti/Nb/Au electrodes on top of them with varying separations as described in the Methods section. The optical image of one such PtTe$_2$ flake (Atomic Force Microscopy image shown in Supplementary Note 1) of around 17.5 nm thickness (33–34 layers) with multiple lateral Josephson junctions of varying separations (L1-L4) is shown in the inset to Fig. 1b which also shows the defined Cartesian coordinate axes for the device. The direction of current bias in these devices is fixed along the $x$-axis. The shortest separation between the niobium electrodes is around 390 nm, and the device is roughly

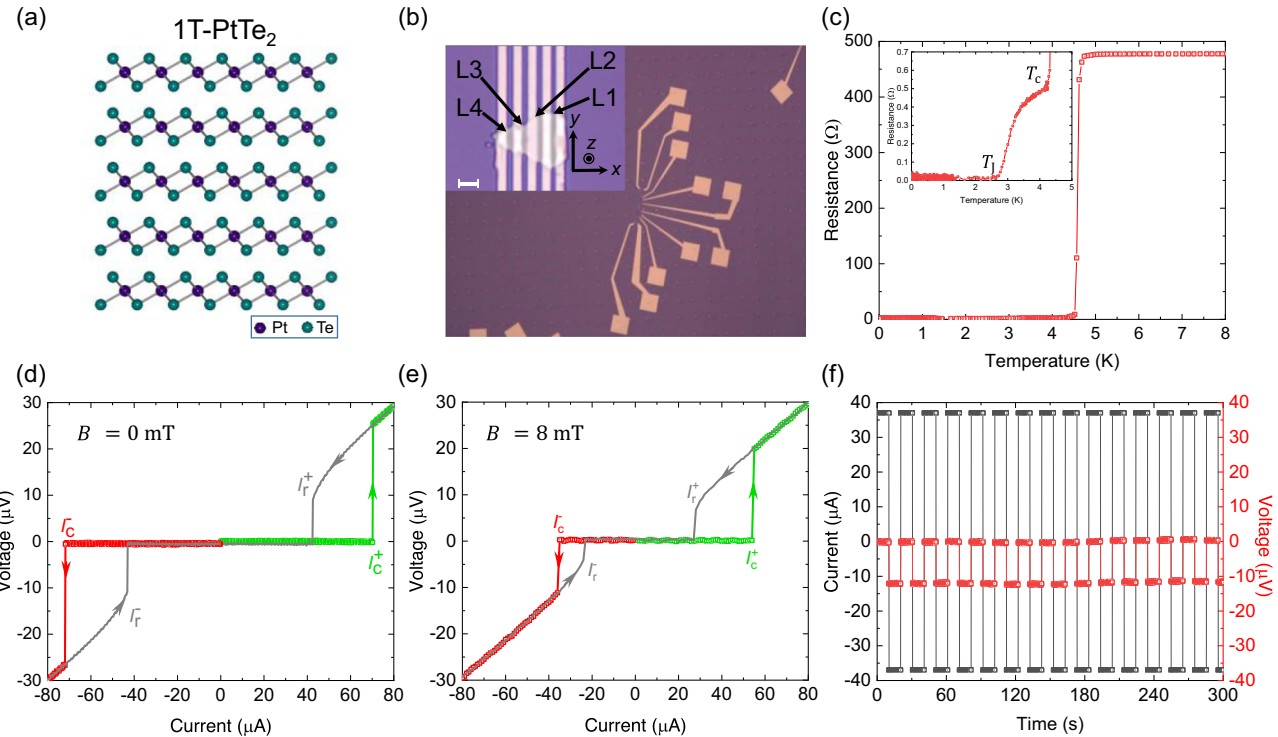

**Fig. 1 | Josephson junctions of PtTe$_2$ and non-reciprocal critical currents. a** A schematic of the 1T-PtTe$_2$ crystal, which shows the structure of five layers of platinum ditelluride with a trigonal prismatic coordination. Purple color indicates platinum atoms and green color indicates tellurium atoms. The structure comprises two dimensional, centrosymmetric layers of PtTe$_2$ stacked on top of each other separated by a Van der Waals gap. **b** An optical image of Josephson junction devices fabricated on a single 17.5 nm thick PtTe$_2$ flake. Inset shows a close up of devices with niobium electrodes with increasing separations labelled L1 to L4. The cartesian coordinate axes that are used are shown. The white scale bar represents 2 μm. **c** Resistance curve of junction L1 measured while cooling down in zero field. Two transitions at around 4.5 K and 2.7 K (inset) corresponding to the superconductivity of niobium ($T_c$) and the junction ($T_J$) are observed. **d** Current-Voltage characteristics of L1 measured in the absence of any external magnetic field after cooling down

in zero magnetic field. The critical currents in the positive ($I_c^+$) and negative ($I_c^-$) directions (in green and red respectively) are the same within the limit of error, making $\triangle I_c = 0$. The retrapping currents in both directions ($I_r^+$ and $I_r^-$) are also equal. **e** Current-Voltage characteristics of L1 measured in the presence of an 8 mT magnetic field applied along y-axis ($B_y$). In addition to a suppression of the energy gap of the junction, we also observe that there is a significant difference in $I_c^+$ and $I_c^-$ leading to a diode effect ($\triangle I_c$). **f** The non-reciprocal behavior of supercurrents measured under the same 8 mT magnetic field with a 37 μA current shows that the device is superconducting along one direction but resistive in the other direction. The switching was measured over a period of one hour and showed robust behavior. Errors in the measurement of voltage arise from the nanovoltmeter or lock-in amplifier and are typically smaller than 40 nV.

5 μm in width. The results presented in the main text are from this device (referred to as L1 hereafter) unless specified otherwise. The resistance of this junction is measured with a small current as the junction is cooled down [Fig. 1c]. A drop in resistance is observed around 4.5 K corresponding to the superconducting transition of the Nb electrodes and another drop at 2.7 K [Fig. 1c inset], below which the junction becomes fully superconducting ($T_J$). The normal state transport characteristics of PtTe$_2$ are provided in Supplementary Note 2.

After cooling the sample to the base temperature of the dilution refrigerator (20 mK), current-voltage curves are measured in zero field. The critical currents on sweeping the current from zero bias in the positive ($I_c^+$) and negative ($I_c^-$) directions are obtained at zero magnetic field [Fig. 1d] and a negligible difference in their magnitude ($\Delta I_c = I_c^+ - |I_c^-|$) or JDE is observed. As the in-plane magnetic field perpendicular to the direction of current ($B_y$) is increased, the appearance of a non-zero $\Delta I_c$ is seen as shown in Fig. 1e. Previously, such a $\Delta I_c$ has also been observed in similar lateral junctions formed with 1T-NiTe$_2$[17], a transition metal dichalcogenide material with the same crystal symmetry and large spin-orbit splitting close to the Fermi level. The JDE has been attributed to the finite momentum Cooper pairing[18] induced by either the topological surface states with large Rashba spin-splitting or the Meissner screening currents within the electrodes.

It is important to note that in contrast to NiTe$_2$ junctions used in previous studies[17], the width ($W$, lateral dimension perpendicular to the direction of current flow) of the PtTe$_2$ flake forming the Josephson junction L1 is comparable to the Josephson penetration depth ($W \sim \lambda_J$). In this limit, the effect of current-induced magnetic field, also known as 'self-field effect' (SFE) becomes significant and the geometry of the current source configuration can play a significant role in the current distribution across the junction. The SFE modifies the critical current of the junction, which can result in skewed Fraunhofer pattern under an out of plane magnetic field. A self-consistent treatment of the wide Josephson junction as described in Supplementary Note 3 is used to simulate the properties of the junction. It is shown in Supplementary Note 4 that in such junctions, it is possible to obtain extraneous JDE just by choosing the current bias electrodes to be on the same side and that this can be avoided by choosing a 'criss-crossed' current bias geometry that gives a rather uniform current distribution across the junction and minimizes the effect of the self-field. Detailed discussion on SFE and the extrinsic JDE resulting from it is provided in Supplementary Note 4. The remarkable match between our experimental data and simulations in both bias configurations reinforces the validity of our supposition. All measurements henceforth, presented on lateral junctions of PtTe$_2$ to determine the spin-momentum locking, were carried out in the criss-crossed geometry to minimize the influence of SFE.

### Helical spin-momentum locking induced JDE in PtTe$_2$

As stated earlier, PtTe$_2$ has helical spin-momentum locked states close to the Fermi level[26]. This helical spin-momentum locking is expected to give rise to a finite-momentum Cooper pairing (FMCP) and a $\Delta I_c$ in the presence of an in-plane magnetic field perpendicular to the current ($B_y$)[17,18]. In the junction geometry used, finite momentum Cooper pairing can also arise due to the Meissner screening currents in the superconducting electrodes[13,17,22,23] but we argue in the discussion section that this effect is negligible and contributes very little to the observed diode effect based on the transparency of our junctions. To establish the presence of helical spin-momentum locking and rule out the presence of three-dimensional spin-orbit coupling in PtTe$_2$, two different configurations of devices were used: lateral junction (L1) as discussed above and a vertical Josephson junction (VJJ) with a PtTe$_2$ flake sandwiched by NbSe$_2$ flakes on top and bottom that is labelled V1 (Refer Supplementary Note 5). The absence of any significant contribution to the JDE due to geometric inversion asymmetry in the shape of the PtTe$_2$ flake is also verified in Supplementary Note 6.

Figure 2a shows $I_c^+$ and $I_c^-$ as a function of $B_y$ in L1 and the corresponding $\Delta I_c$ is shown in Fig. 2b. It can be seen that $\Delta I_c$ increases linearly with $B_y$ at low fields and then starts to fluctuate and decrease non-monotonously. This is due to an additional magnetic flux to the sample from

$B_y$ that can either be due to the finite sample thickness or a tiny misalignment or flux focusing effect that leads to an additional phase difference across the electrodes. For this reason, we use the Fraunhofer interference pattern to locate the exact critical current maxima where the net out-of-plane magnetic flux is zero and extract the value of the diode effect due to $B_y$ only (Refer Supplementary Note 7). This method of extracting the diode effect helps avoid any pitfalls due to magnetic flux through the sample. The effect of flux focusing and the determination of the flux focusing factor (Γ) is described in Supplementary Note 8.

On measuring the critical currents $I_c^+$ and $I_c^-$ of L1 at 20 mK as a function of the in-plane magnetic field angle [Fig. 2c], it is seen that $\Delta I_c$ is maximized when the magnetic field is applied perpendicular to the direction of current ($B_y$) and vanishes when the magnetic field is along the direction of current ($B_x$). $\Delta I_c$ also decreases as a function of temperature at higher temperatures with a quadratic $(T - T_J)^2$ dependence as expected for a finite momentum Cooper pairing scenario[17,18], as shown in Fig. 2d. The angular dependence of $\Delta I_c$ with increasing temperature is shown in Fig. 2e and the interference pattern in the forward and backward directions measured using a criss-crossed geometry as a function of the z-axis magnetic field ($B_z$) is shown in Fig. 2f. There is no diode effect observed for a magnetic field along the z-axis. Furthermore, there is no clear evidence of a $\Delta I_c$ in vertical Josephson junctions of PtTe$_2$ with an in-plane magnetic field along different directions (Refer Supplementary Note 5), as opposed to that in vertical junctions of T$_d$-WTe$_2$ where a clear $\Delta I_c$ is observed[31]. Thus, this result shows the absence of net spin-momentum locking or any other finite momentum pairing mechanism when the current flows along the c-axis of PtTe$_2$. All the above results together point to the existence of a two-dimensional helical spin-momentum locking in PtTe$_2$.

### Tunable second-order supercurrents and Current-Phase relationship (CPR) induced by Finite Momentum Cooper Pairing (FMCP) in PtTe$_2$

Having established the existence of a helical spin-momentum locking in PtTe$_2$, the evolution of the Fraunhofer pattern in lateral PtTe$_2$ junctions in the presence of $\Delta I_c$ is studied to gain insight into the CPR of the system. While superconducting quantum interference devices (SQUIDs) are the preferred platforms to deduce the current-phase relationship in a system, Josephson junctions have the advantage that the distribution of super-currents in the system may also be obtained by analyzing the Fourier transform of the Fraunhofer pattern. The Fraunhofer patterns for the critical currents, $I_c^+$ and $I_c^-$ are measured as the function of the magnetic flux Φ along the z-direction, under various $B_y$ is shown in Fig. 3a–d after correcting for flux focusing effects[32,33] and the finite thickness effect[34,35] of the sample as discussed in detail in Supplementary Notes 7 and 8. When $B_y = 0$, $I_c^+$ and $I_c^-$ lie on top of each other leading to a negligible $\Delta I_c$ and the period of oscillations is close to a single magnetic flux quantum ($\Phi_0 = \frac{h}{2e}$) as expected [Fig. 2f]. As $B_y$ is increased in the negative direction to -8 mT and the Fraunhofer pattern is measured again [Fig. 3a], it is observed that the central maxima of $I_c^-$ increases slightly in magnitude while the magnitude of the central peak of $I_c^+$ starts to decrease. As the magnetic field is increased further from -12 mT to -24 mT, [Fig. 3b–d] the central peak of $I_c^-$ doesn't decrease much in magnitude while the magnitude of the central peak of $I_c^+$ has a sharp decrease in the middle leading to the formation of a sharp noticeable dip in critical current where maximum $\Delta I_c$ is observed. It is to be noted that in these experiments, the roles of $I_c^+$ and $I_c^-$ are reversed when $B_y$ is swept in the opposite direction and that corresponds to $I_c^+(B_y, B_z) = -I_c^-(-B_y, -B_z)$, indicating that the total time-reversal symmetry of the system is maintained and there is no other external sources of magnetic flux, like vortices trapped in the system (Refer Supplementary Note 9). The details of $I_c^+$ and $I_c^-$ measured in the other junctions (L3 and L4) are presented in Supplementary Notes 10–12.

The appearance of $\Delta I_c$ in PtTe$_2$ can be understood using a simple model starting from a general current-phase relationship (CPR) written as a Fourier series of sine functions, which includes higher harmonics and additional phase shifts $\varphi_n$ that may be present in the system when time-

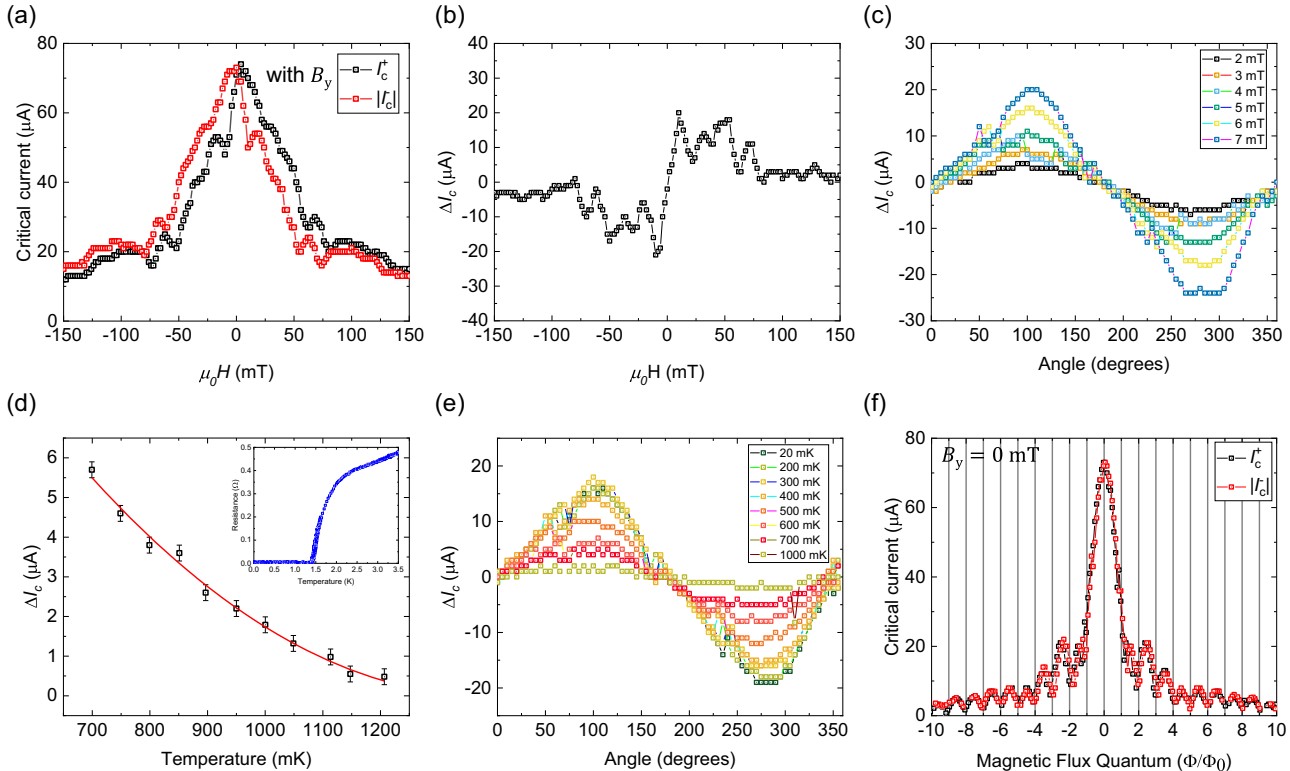

**Fig. 2 | Analysis of $\Delta I_c$ with magnetic field magnitude, angle and temperature for junction L1. a** Positive critical currents $I_c^+$ (in black) and negative critical currents $\left|I_c^-\right|$ (in red) measured as a function of the y-axis magnetic field ($B_y$) swept from 150 mT to −150 mT at 20 mK temperature. **b** Josephson diode effect $\left(\triangle I_c\right)$ simply measured by sweeping $B_y$ from 150 mT to −150 mT, shows that it is maximum around 10 mT and displays non-sinusoidal behavior. **c** The angular dependence of $\triangle I_c$ at various magnetic fields measured at 20 mK shows that $\triangle I_c$ is maximized when the magnetic field is perpendicular to the direction of current and zero when the magnetic field is parallel to the direction of current indicating a helical spin-momentum locking in the system. **d** This figure shows the temperature dependence

of $\triangle I_c$ measured in device B1 at higher temperatures and $B_y = 24$ mT. The fit represents a quadratic $(T − T_J)^2$ dependence with the junction transition temperature $T_J \approx 1.4$ K. Inset shows the measurement of resistance vs temperature in the presence of a magnetic field $B_y = 24$ mT, which shows $T_J$ to be around 1.4 K (**e**) The angular dependence of $\triangle I_c$ with the magnetic field $B_y = 8$ mT measured at various temperatures. **f** $I_c^+$ and $I_c^-$ measured as a function of the z-axis magnetic field ($B_z$) measured in a criss-crossed configuration shows negligible $\triangle I_c$. The critical current is defined as the current at which the derivative of voltage with respect to the current exceeds a specific threshold. The error in the detection of critical current is determined by the step size of the current sweep, which is around 250 nA.

reversal symmetry is broken.

$$I(\varphi) = \sum_{n=1}^{\infty} I_n \sin(n\varphi + \varphi_n) \qquad (1)$$

This CPR can be expanded up to the second order as higher order supercurrents contribute negligibly to the total current. This gives us:

$$I(\varphi) = I_1 \sin(\varphi + \varphi_1) + I_2 \sin(2\varphi + \varphi_2) \qquad (2)$$

Certain well-known cases of unconventional CPR can be derived from this generic CPR. For example, having $\varphi_1 = \varphi_2 = 0$ in Eq. 2 gives a CPR that contains only the first and second harmonic terms without any additional phases corresponding to typical $\varphi$-junctions[36,37] with a skewed current-phase relationship. In the case where $I_2 = 0$ in Eq. (2), it gives rise to anomalous Josephson junctions or $\varphi_0$-junctions with a sinusoidal current phase relationship shifted from zero by a phase $\varphi_1$. Such CPRs have been observed typically in ferromagnetic Josephson junctions and systems with high spin-orbit coupling[38–43]. Now, without any loss of generality $\varphi$ may be replaced with $(\varphi - \varphi_1)$ and the CPR can be rewritten as:

$$I(\varphi) = I_1 \sin\varphi + I_2 \sin(2\varphi + \delta) \qquad (3)$$

introducing $\delta = \varphi_2 - 2\varphi_1$, the relative phase between the first and second harmonic terms. We shall call such a Josephson junction with a CPR as that in Eq. (3) as a 'tunable second order $\varphi_0$-junction'. This CPR is identical to

that derived from the Ginzburg-Landau formalism[17], in which $\delta$ corresponds to the phase shift induced by a finite momentum Cooper pairing in the system. The phase, $\delta$, may be controlled by an in-plane Zeeman field perpendicular to the direction of current ($\delta \propto B_y$). We note that, in addition to NiTe$_2$[17], similar CPRs have been used to explain the presence of a $\Delta I_c$ in InAs-based superconducting junctions[15,44] and InSb nanowire junctions[45].

$\Delta I_c$ may be determined by examining $I_c^+$ and $I_c^-$ from the CPR in Eq. (3). For non-sinusoidal CPRs, such as that in Eq. (3), that are composed of higher order Fourier harmonics, critical currents may not occur at $\varphi = \pm\frac{\pi}{2}$ and need to be solved numerically to obtain the exact values of $I_c^+$ and $I_c^-$ for different values of $\delta$. We obtain the critical currents as

$$I_c^+(\Phi, \delta) = \max_{\varphi}[I_{tot}(\varphi, \Phi, \delta)]$$

$$I_c^-(\Phi, \delta) = \min_{\varphi}[I_{tot}(\varphi, \Phi, \delta)]$$

where $I_{tot}(\varphi, \Phi, \delta)$ denotes the total current given by

$$I_{tot}(\varphi, \Phi, \delta) = \frac{1}{W} \int_{-\frac{W}{2}}^{\frac{W}{2}} dy\, I\left(\varphi + 2\pi\frac{\Phi}{\Phi_0}\frac{y}{W}\right)$$
$$= \left(I_1 \sin\varphi + I_2 \cos\left(\pi\frac{\Phi}{\Phi_0}\right)\sin(2\varphi + \delta)\right)\frac{\sin(\pi\Phi/\Phi_0)}{\pi\Phi/\Phi_0}$$

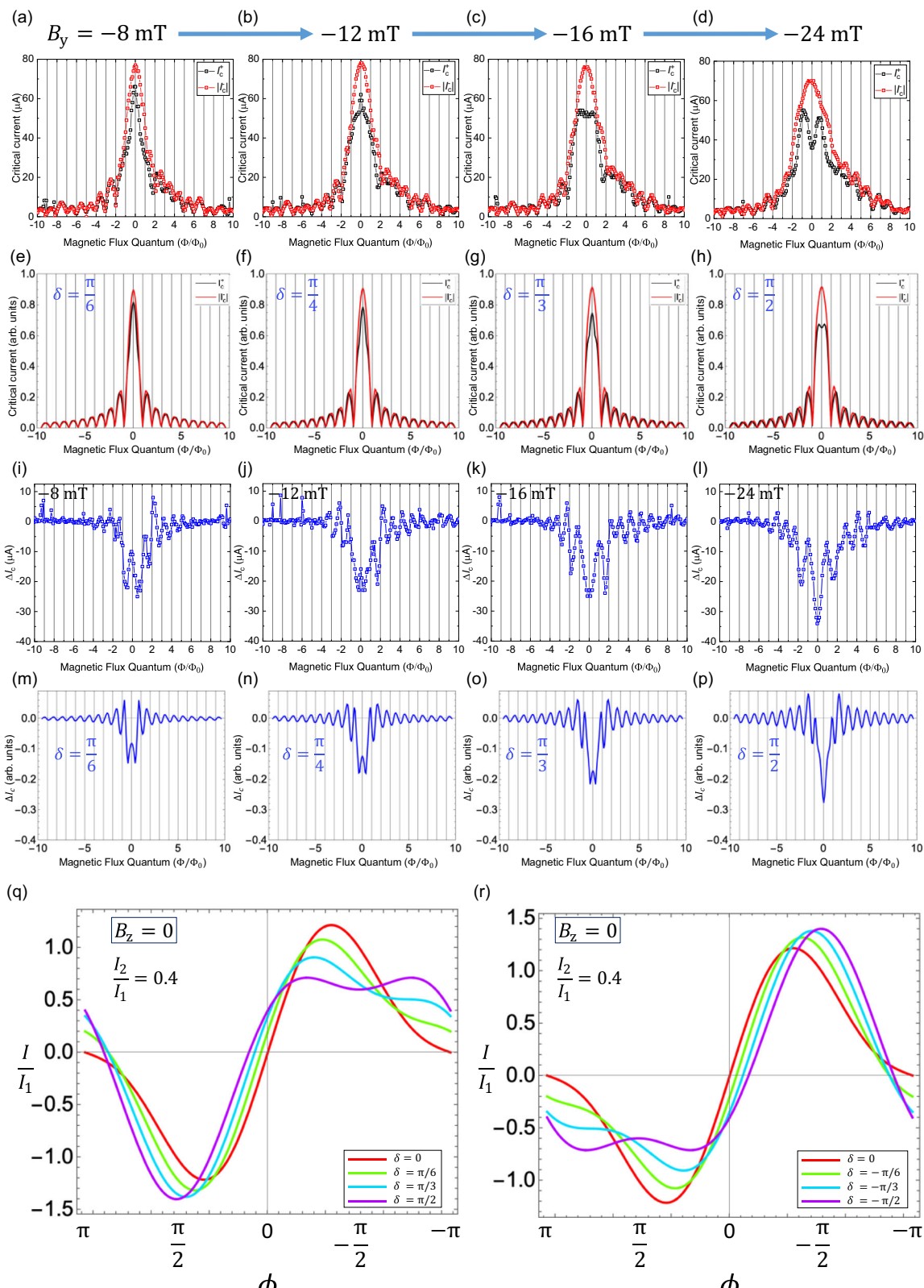

For $\frac{I_2}{I_1} \ll 1$, the critical currents $I_c^{\pm}$ occur close to $\varphi = \pm \frac{\pi}{2}$, so that

$$\Delta I_c(\Phi, \delta) = -2I_2(B)\sin\delta \left[\frac{\sin(\frac{2\pi\Phi}{\Phi_0})}{\frac{2\pi\Phi}{\Phi_0}}\right] \quad (4)$$

where $I_n(B) = I_n(0)\left(1 - \frac{B^2}{B_c^2}\right)^n$ accounts for the suppression in the critical current components due to $B$. We note that $\Delta I_c$ can serve as a probe for second harmonic supercurrent in the junction because the first harmonic term does not have any direct contribution to $\Delta I_c$. From Eq. (4), we infer three important conclusions on the nature of the CPR and $\Delta I_c$. First, we find

**Fig. 3 | Evolution of the Fraunhofer pattern in the presence of $\Delta I_c$ for junction L1.** **a–d** Shows the experimentally measured Fraunhofer patterns for positive and negative critical currents $I_c^+$ and $I_c^-$ in the presence of a negative magnetic field along y-axis ($B_y$) of different magnitudes up to $-24$ mT. A $0-\pi$ junction-like dip is observed in $I_c^+$ upon increasing the magnitude of $B_y$. **e–h** shows the simulated Fraunhofer patterns using a self-consistent treatment (as described in Supplementary Note 3) for the ratio of second and first harmonic supercurrents $\frac{I_2}{I_1} = 0.4$. The behavior of $I_c^+$ and $I_c^-$ with increasing the relative phase difference ($\delta$) in the simulations is similar to that observed in experiment with increasing $B_y$. **i–l** shows the increasing diode effect ($\Delta I_c$) with the increasing magnitude of $B_y$ reaching maximum value around $-24$ mT. **m–p** Simulated $\Delta I_c$ using the current-phase relationship (CPR) in Eq. (3) for similar magnetic fields as in experiment. The experimentally observed features including the dip are captured well by the simulation. **q, r** shows the CPRs corresponding to negative and positive $B_y$ used in the simulations for $\frac{I_2}{I_1} = 0.4$. The non-reciprocal response of $I_c^+$ and $I_c^-$ under $|B_y|$ is evident from these simulations. The error in the detection of critical current is determined by the current step size, which is around 500 nA.

that the existence of a second-harmonic term ($I_2 \neq 0$) and $\delta \neq n\pi$, $n \in \mathbb{Z}$ is necessary for the existence of a non-zero $\Delta I_c$. So, the presence of a $\Delta I_c$ acts as an indicator for the existence of a second harmonic term in the current-phase relationship while the converse is not true. Second, control over $\delta$ leads to the possibility of tuning the relative phase between the first and second harmonic components, which leads to control of specific harmonics. For example, substituting $\delta = \pi$ in Eq. (3) gives:

$$I(\varphi)_{\delta=\pi} = I_1 \sin\varphi + I_2 \sin(2\varphi + \pi) = I_1 \sin\varphi - I_2 \sin 2\varphi \quad (5)$$

In this CPR, the first and second harmonics of supercurrents have opposite signs and can flow in opposite directions. Hence by tuning the magnetic flux and choosing a suitable value of $\delta$ and $\varphi$, the magnitude and flow direction of pure second or first order supercurrents across the junction can be controlled. Third, the magnitude of $\Delta I_c$ is modulated by $\sin\delta$, which implies that $\Delta I_c$ reaches its largest magnitude when $\delta \approx \pm\frac{\pi}{2}$.

In order to corroborate the validity of the model, the Fraunhofer patterns simulated for $I_c^+$, $I_c^-$ based on this CPR is presented in Fig. 3e–h. The value of $\frac{I_2}{I_1}$ used in these simulations is calculated as presented below. The experimentally obtained $\Delta I_c$ for different $B_y$ is presented in Fig. 3i–l. In a system with FMCP such as 1T-PtTe$_2$, $\delta$ may be tuned precisely with an in-plane magnetic field ($B_y$). Assuming that $I_2$ and $I_1$ are both positive, the value of $B_y$ at which $\Delta I_c$ reaches the maximum (minimum) value $\Delta I_c^{max}$ ($\Delta I_c^{min}$) corresponds roughly to $\delta = -\frac{\pi}{2}$ ($\delta = \frac{\pi}{2}$) from Eq. (4). Using the value of $\Delta I_c^{min}$ in Eq. (4), we observe that the magnitude of second harmonic supercurrent flowing through the junction is $I_2(B_y) = -\frac{\Delta I_c^{min}}{2}$, in the limit of $\Phi$ going to zero. For junction L1, the minimum value of $\Delta I_c$ is around $-34\,\mu A$ at $B_y = -24$ mT [Fig. 3l], this would produce $I_2(-24\,\text{mT}) \approx 17\,\mu A$ and the actual value of $\frac{I_2(0)}{I_1(0)} \approx 0.4$. The value of $\frac{I_2}{I_1}$ obtained from this analysis is larger than that measured in some semiconductor junctions with high transparency such as Sn-InSb nanowire junctions[45] and comparable to that observed in Al-InAs planar Josephson junctions[46].

$\Delta I_c$ corresponding to different values of $\delta$ are simulated with $\frac{I_2}{I_1} \approx 0.4$ and are presented in Fig. 3m–p. We observe that the CPR captures the main features of the experimental data such as the magnitude of $\Delta I_c$ and the oscillation period of it. Some additional features for small $B_z$ such as lifted nodes in $\Delta I_c$ and the formation of a dip in the critical currents at zero $B_z$ can be captured by introducing an additional term to the phase difference such that $\varphi \to \varphi + \beta \frac{I_{tot}}{I_1} |\frac{y}{W}|$, which is due to a small remnant self-field in the junction. In Supplementary Note 3, we provide a derivation of $\beta$ and show that $\beta = \frac{1}{2} \left(\frac{W}{\lambda_J}\right)^2$. This term directly shows the influence of junction geometry (wide junction) on the phase gradient. The corresponding CPRs for negative and positive $B_y$ are shown in Fig. 3q, r respectively, where the non-reciprocal nature of the critical currents can be seen clearly. The details of the simulations are relegated to Supplementary Note 3. It is seen that the features of $I_c^+$ and $I_c^-$ from the simulation are in qualitative agreement with the experimentally measured curves. We note that the features of the simulation that are also observed in experiment such as the sharp peak in $I_c^+$ around -12 mT and the observed dip in $I_c^+$ beyond -16 mT are quite sensitive to the value of $\frac{I_2}{I_1}$ and the origin of these features are reflected in the calculated CPR

curves [Fig. 3q]. It can be seen in the CPR curves that as $|B_y|$ is increased, the critical current in the negative direction ($I_c^-$) first increases in magnitude and then starts to decrease gradually with a steady shift in the value of $\varphi$ at which it occurs, while in the case of $I_c^+$ there is initially a gradual decrease in its value as $B_y$ is increased with a shift in the value of $\varphi$.

Further, the evolution of $I_c^+$ and $I_c^-$ as a function of $B_y$ plotted in Fig. 4a is well replicated by the corresponding simulation presented in Fig. 4b. The absence of nodes in the experimental observation of $\Delta I_c$ versus $\delta$ in Fig. 4a and c can be explained by the presence of a small magnetic flux induced by $B_y$ as shown in Supplementary Note 13. At this point, a digression on the effect of the in-plane magnetic flux due to $B_y$ is warranted. The effective in-plane cross sectional area of the junction including the London penetration depth of the two superconducting electrodes is around $1.1725 \times 10^{-14}\,\text{m}^2$, which gives the effective magnetic field needed to induce a single magnetic flux quantum in the junction is around 176 mT and the in-plane fields that we use in our study is much lower than this (0–50 mT) to create any magnetic flux-induced oscillations. Moreover, the evolution of $I_c$ with $\delta$ is quite different from what is expected for a typical flux-induced Fraunhofer pattern. There is an inherent asymmetry between $I_c^+$ and $I_c^-$ that arises due to $\delta$ as shown by the simulations in Fig. 4b and observed experimentally in Fig. 4a. Rather than having nodes, the evolution of $I_c$ with $\delta$ has oscillations that decay slowly with no nodes in the critical current. Hence, we conclude that flux-induced Fraunhofer interference effects due to $B_y$ in our junctions are not relevant to the observed diode effect.

The intimate correlation between the experimentally observed features [Fig. 4a] and the simulation [Fig. 4b] demonstrates the accuracy of the assumed CPR. Figure 4c shows $\Delta I_c$ as a function of $B_y$ as derived from the Fraunhofer interference pattern at zero net magnetic flux. $\Delta I_c$ deviates from the expected sinusoidal behavior and increases in magnitude linearly with $B_y$ till $\pm 24$ mT and then decreases linearly towards zero. This behavior can also be reproduced successfully in the simulations by tuning the $|\frac{I_2}{I_1}|$ ratio as shown in Fig. 4d. While $\Delta I_c$ vs $B_y$ remains sinusoidal for lower values of $|\frac{I_2}{I_1}|$, it gradually turns triangular for larger values of $|\frac{I_2}{I_1}|$ for wide junctions. This deviation of $\Delta I_c$ from the sinusoidal behavior expected from Eq. (4) also confirms the presence of large second-harmonic supercurrents.

One of the main observations from Eq. (4) is that $\Delta I_c$ is expected to oscillate with the magnetic flux $\Phi$ with nodes at every half-flux quantum $(\frac{\Phi_0}{2})$ due to the presence of the second-harmonic term in the CPR. The oscillations in $\Delta I_c$ as a function of $\Phi$ at $B_y = 20$ mT are presented in Fig. 4e. Though the oscillations are expected to vanish, the first few nodes in $\Delta I_c$ are lifted from their zero position, which is similar to that observed in the Fraunhofer patterns for $I_c^+$ and $I_c^-$ [Fig. 2f]. Lifted nodes in Fraunhofer patterns can occur due to several mechanisms, such as the junction geometry[47], current asymmetry, or a remnant of an unconventional CPR due to topological superconductivity[48,49]. The lifted nodes encountered in our case can be accounted for in the simulations by the presence of a self-field related to the junction geometry that results in non-zero $\beta$. $I_c^+$ is simulated for different values of $\beta$. It can be seen that the appearance of a dip in $I_c^+$ upon increasing $\delta$ [Fig. 4f] can also be captured by increasing $\beta$ [Fig. 4g], suggesting the intimate correlation between these two parameters as assumed. The variations in the magnetic flux at which these features can be observed experimentally is due to a varying flux-focusing factor close to zero magnetic flux[33] and the first lifted node in $I_c^+$, which masks the dip close to the first magnetic flux quantum. The simulation of lifted nodes in $\Delta I_c$ due

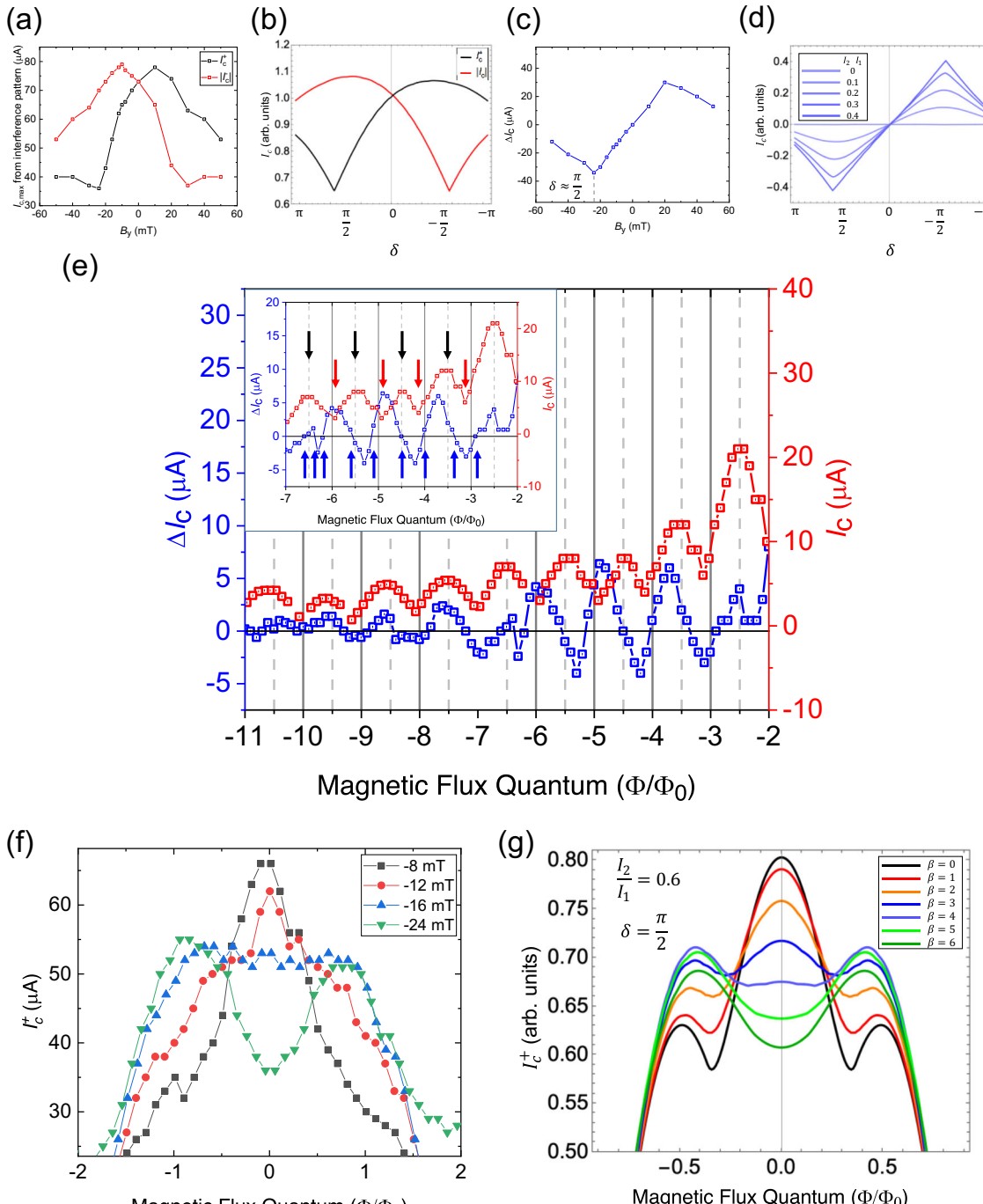

**Fig. 4 | Evolution of the diode effect ($\Delta I_c$) with the phase difference between the harmonics ($\delta$) and magnetic flux in junction L1. a** The evolution of positive (in black) and negative (in red) critical currents, $I_c^+$ and $I_c^-$ of the central maxima in the Fraunhofer pattern at positive and negative magnetic field along y-axis ($B_y$) after correcting for finite thickness shifts. **b** Simulated evolution of $I_c^+$ and $I_c^-$ using the current-phase relationship in Eq. (3). It is found to fairly replicate the experimentally observed features. The absence of nodes in (**a**) can be replicated by the presence of additional magnetic flux as shown in Supplementary Note 13. **c** $\Delta I_c$ from the Fraunhofer patterns calculated after correcting for finite thickness effects in the junction. The minimum in $\Delta I_c$ is expected to occur close to $\delta \approx \frac{\pi}{2}$. **d** $\Delta I_c$ calculated from the simulated Fraunhofer patterns for a wide junction, where $|\frac{I_2}{I_1}|$ increases in increments of 0.1. The maxima (minima) deviates from $\delta = -\frac{\pi}{2}$ ($\frac{\pi}{2}$) with increasing values of $|\frac{I_2}{I_1}|$. $\Delta I_c$ evolves from a sinusoidal dependence at low values of $|\frac{I_2}{I_1}|$ to a nearly triangular behavior at higher values of $|\frac{I_2}{I_1}|$. **e** The evolution of $\Delta I_c$(in blue) and $I_c$(in red) with magnetic flux $\Phi$ with $B_y = 20$ mT. Inset shows a close up of

oscillations in $\Delta I_c$ has nodes appearing roughly at half magnetic flux quantum $\left(\frac{\Phi_0}{2}\right)$ frequency(denoted by blue arrows) and has almost double the frequency compared to the nodes in the critical current (denoted by red arrows) that happens roughly at every magnetic flux quantum ($\Phi_0$). It is clearly noticeable that there are twice as many blue arrows compared to red arrows. Black arrows indicate the position of the antinodes in $I_c$. The position of the nodes are slightly altered from half magnetic flux quantum due to the presence of lifted nodes and varying flux focusing factor with increasing magnetic flux. **f** The experimental evolution of $I_c^+$ with $\Phi$ for different values of $\delta$ shows the appearance of a dip with increasing $\delta(-B_y)$. **g** Similar appearance of a dip-like feature in $I_c^+$ is captured in the simulations by tuning the parameter $\beta$. The difference in the values of magnetic flux between the experiment and the simulations can occur due to the presence of a variable flux-focusing factor[33]. The error in the detection of critical current is determined by the current step size, which is around 500 nA.

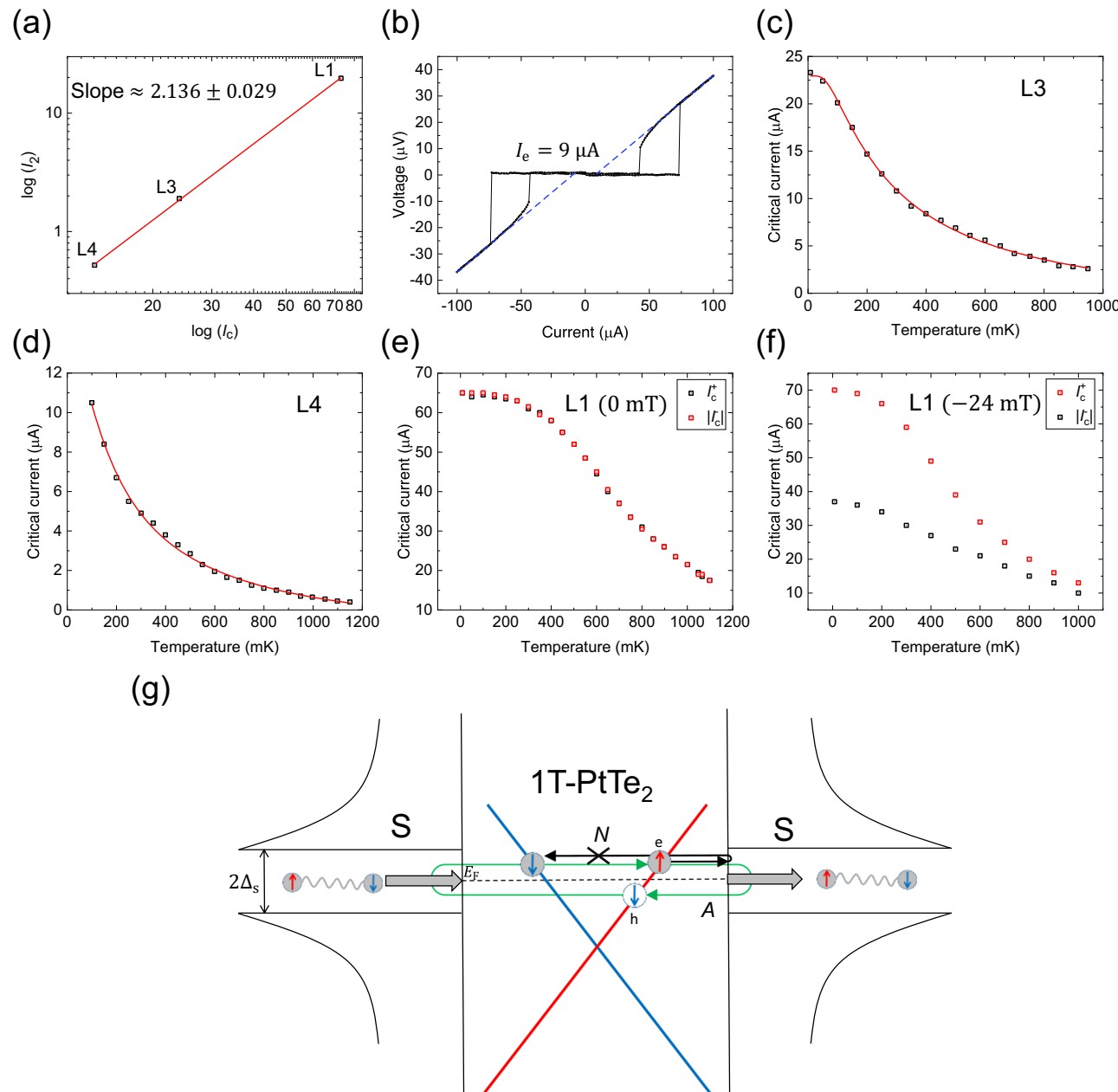

**Fig. 5 | Transparency of PtTe₂ Josephson junctions. a** The log-log plot of the evolution of the second harmonic supercurrent $I_2$ extracted from the diode effect $\triangle I_c$, for different $I_c$ in junctions L1, L3 and L4 shows that they scale quadratically (slope ~ 2) with the critical current as expected ($I_2 \propto I_c^2$, for $I_2 \ll I_1$), further validating our assumed current-phase relationship. **b** The current-voltage curve for L1 junction measured at 20 mK shows the presence of excess currents around 9 μA (determined by extrapolating the linear voltage regime, represented by blue dashed lines) indicating coherent transport across the junction and a transparency of around 0.45 derived from the OTBK model. **c, d** The critical currents as a function of temperature $I_c(T)$ for junctions L3 and L4 are fit with an equation corresponding to the long junction limit yielding a transparency of around 0.436 and 0.428 respectively. **e** $I_c(T)$ for junction L1 with $I_c$ starting to saturate below 500 mK. There is no significant difference between positive $I_c^+$ (in black) and negative critical currents $I_c^-$ (in red) in the absence of the magnetic field along y-axis ($B_y$). (**f**) $I_c(T)$ for positive (in black) and negative critical currents (in red) $I_c^+$ and $I_c^-$ for magnetic field along y-axis $B_y = -24$ mT for L1 shows that at low temperatures, the critical currents for the negative direction is much larger in comparison with the critical currents for the positive direction indicating that the induced energy gap in the Josephson junction is largely anisotropic in the presence of $B_y$. **g** Schematic of Dirac-cone like helical spin-momentum locking in PtTe₂ that potentially suppresses the normal reflections in the junction due to the reduced availability of spin states. The red and blue colors depict opposite spin polarizations in the band. Green arrow represents the Andreev reflection process (A) and black arrow represents the normal reflection process (N), which is potentially suppressed due to the reduced availability of a state with the same spin. This phenomenon can increase the transparency of the junction, while simultaneously enhancing the phase-coherent Andreev processes, leading to higher harmonics in the supercurrent. The error in the detection of critical current is determined by the current step size, which is around 500 nA.

to a non-zero $\beta$ are presented in Supplementary Note 13. The oscillations are observed to have nodes that are roughly spaced every $(\frac{\Phi_0}{2})$ strongly indicating that the major component in $\Delta I_c$ is close to second harmonic as expected in a tunable second order $\varphi_0$-junction. The second harmonic component extracted from other junctions (L3 and L4) together with L1

(shown in Supplementary Notes 10–12) is observed to scale quadratically with the $I_c$ of the junctions, as shown in Fig. 5a, which is expected from a Ginzburg-Landau analysis[17], further confirming our hypothesis on the direct correlation between the diode effect and the second harmonic supercurrents. The junction L2 that is accidentally shorted by another flake

and forms an asymmetric SQUID displays skewed, non-sinusoidal oscillations of the critical current, also indicating the presence of a large second-harmonic term in the CPR of the junction[50] (Refer Supplementary Note 14). Calculating the Josephson diode efficiency $\eta = \frac{\Delta I_c}{I_c^+ + |I_c^-|}$ for this junction at maximum $\Delta I_c$ gives a value of around 32 % at $-24$ mT, which is one of the largest values reported so far (Refer Supplementary Note 15 for comparison).

## Discussion

Now, we turn our attention to the physical origin of a large second-harmonic term in PtTe$_2$ Josephson junctions. The presence of such a large second harmonic component in a Josephson junction with a large electron density made over such long separations is quite unanticipated as it indicates a large transparency of the interface between the superconducting electrodes and PtTe$_2$, and the presence of resonant Andreev bound states due to phase coherent transport across the junction[51]. The band structure studies of multilayer PtTe$_2$ have shown the formation of high mobility Fermi pockets at the Fermi level[52,53]. This means that the difference between the Fermi level and the edges of these bands is small. As a result, a relatively large external magnetic field can lead to significant warping and even the (dis)appearance of the Fermi pockets, i.e., a Lifshitz transition[54], which could strongly affect the density of states and velocity at the Fermi level. A large density of states available for Cooper pair transfer, as a result, can enhance the critical currents associated with single ($I_1$) and double Cooper pair transfer ($I_2$) in the junction.

Moreover, we note that large transparencies and consequently higher harmonics are observed in high-mobility semiconducting[55] and semi-metallic junctions[56] with low electron densities and pristine interfaces but it is not so in metallic junctions due to short mean-free paths and scattering at the interface. The mean-free path ($l_e$) reported in literature for single crystals of PtTe$_2$ is around 180 nm in the *ab* plane[57], which is large in comparison with a normal metal with similar carrier densities, where it is typically of the order of $1 - 50$ nm[58]. Josephson junctions made with metallic barriers have large critical currents owing to large density of states available for Cooper pair tunneling. As discussed earlier, since the second harmonic supercurrent ($I_2$) scales roughly as the square of $I_c$, $I_2$ is typically larger in metallic Josephson junctions with larger $I_c$ as in our case ($\sim 73$ μA) as compared to semiconductor Josephson junctions with much lower electron densities[16] ($\sim 1$ μA). This enhancement in $I_c$ is advantageous for easier and clear observation of higher order effects in the junction like the oscillations in $\Delta I_c$ that we observe.

The superconducting coherence length $\left( \xi = \frac{\hbar v_F}{\pi \Delta_0} \right)$ of L1 is calculated to be around 200 nm at zero temperature using the average value of $v_F \approx 3.3 \times 10^5$ ms$^{-1}$ reported in literature[57] for PtTe$_2$. It is neither clearly in the short or long junction limit when compared to the junction separation (390 nm) and thus is not straight forward to determine the transparency of the junction by fitting $I_c(T)$ using a standard model for a long junction. Instead, the transparency of this junction is obtained by examining excess currents ($I_e$) that are obtained by linear extrapolation of the $I - V$ curve above the critical current back to zero voltage, as shown in Fig. 5b. The existence of $I_e$ in a highly transparent junction with long-range phase-coherent Andreev reflections is explained by the Octavio-Tinkham-Blonder-Klapwijk (OTBK) model[59,60]. The $I_e$ for L1 at 20$mK$ is around 9 μA which corresponds to a transparency of around 0.45.

The transparency of the other junctions L3 and L4, which are in the long junction limit ($d \gg \xi$), is obtained by fitting the critical current over the entire temperature range in the long junction limit[61,62] given by $I_c(T) = \eta \frac{aE_T}{eR_n} \left[ 1 - be^{\frac{-aE_T}{3.2k_BT}} \right]$. $a$ and $b$ are fitting parameters. $E_T$ is the Thouless energy and $R_n$ is the normal state resistance. The details of the fit can be found in the SI. The extracted transparency from the fits for L3 and L4 [Fig. 5c, d] is around 0.436 and 0.428 respectively, which is consistent with the values from excess currents. Figure 5e, f show the critical currents in L1 with temperature in the absence and presence of $B_y$ respectively,

demonstrating the strong anisotropy in the critical currents that develops in the presence of $B_y$. The transparency of the junctions is not as large as that found in semiconductor junctions, which is closer to unity in ballistic transport. In the case of PtTe$_2$, the decreased transparency is due to the contribution of diffusive channels in parallel to transparent ballistic channels. The presence of relatively high transparency despite the ex-situ fabrication of junction interface can be attributed to the significant contribution of the states with helical spin-momentum locking to the transport of supercurrents. We conjecture that the states in PtTe$_2$ with a spin-momentum locked Dirac-like dispersion[26] can suppress normal reflections due to the reduced availability of spin conserving states. This leads to coherent Andreev processes over long distances leading to strong second harmonic supercurrents as has been reported previously in other topological systems[63–65] and depicted in Fig. 5g. For instance, an electron in PtTe$_2$ moving to the right with an up spin, incident on the surface of the superconductor can be reflected as a hole moving to the left into the same band with opposite spin, which corresponds to the Andreev reflection process. Whereas for the normal reflection process which conserves spin upon reflection, the availability of spin states for the reflected electron is strongly dependent on the incident angle, in a system with spin-momentum locked Dirac-like dispersion as shown previously[63] and can be highly suppressed for incidences close to the normal. Similar arguments have also been presented in another recent work on highly transparent Dirac semimetal MoTe$_2$ junctions[66].

The transparency of the junctions also provide important clues into the dominant mechanism of JDE in our junctions and determining the effect of trivial mechanisms such as Meissner screening currents[13,17,22,23] that can also lead to a JDE. Application of a magnetic field perpendicular to current can induce finite-momentum Cooper pairing in materials with helical spin-momentum locking. Finite momentum Cooper pairs can also be generated through Meissner screening currents in the superconducting electrodes that can also induce a JDE[13,22,23]. While it is hard to completely disentangle the JDE arising due to these two mechanisms, we can make arguments based on the junction transparency to establish the dominant role of helical spin-momentum locking in the observation of a large JDE. It is important to note that the JDE generated by Meissner screening currents is extremely sensitive to the transparency of the junction[13,67] and drops drastically with decrease in transparency. For junctions with transparency of 0.45, the maximum diode efficiency due to Meissner screening currents is predicted to be around 4%[13,67]. In our case, the experimentally observed diode efficiency is close to 32% for similar transparencies, which is much larger and cannot be accounted for completely by finite momentum Cooper pairs created by orbital effects only.

The large spin-orbit coupling effect at small magnetic fields as evidenced from the non-zero momentum of the Cooper pairs indicates a large $\left( \frac{g}{v_F} \right)$ ratio in PtTe$_2$, which in turn points to the presence of an extremely large $g$-factor (in the order of $10 - 100$) for the electrons in PtTe$_2$ that depends on the exact value of $v_F$ for the bands that contribute to the supercurrent transport. However, it is hard to precisely estimate the $g$-factor directly from the JDE. A discussion on the $g$-factor estimation from the JDE and its limitations are presented in Supplementary Note 16. It is to be noted that similarly large $g$-factors have been previously reported in other topological semimetals[68,69] and semiconductor heterostructures[70,71]. The large spin-orbit coupling and Zeeman splitting with small magnetic fields coupled with the strongly coherent higher order Cooper pair transport in PtTe$_2$, as evidenced by its large JDE, provides an interesting alternative platform to engineer topological superconductivity in planar Josephson junctions[72,73] as has been demonstrated before in Josephson junctions of HgTe[74] and InAs[75] quantum well structures. One of the major challenges in the current existing platforms for realizing topological superconductivity is the engineering of high quality interfaces[76,77]. The complete air stability of PtTe$_2$ and the states with strong helical spin-momentum locking allow for creation of high quality interfaces with superconductors without many complications.

## Conclusion

In summary, we have shown through measurements of $\Delta I_c$ that the Dirac semimetal 1T-PtTe$_2$ has a large JDE that arises from its helical spin-momentum locked states under a Zeeman field. While extrinsic effects such as SFE can be present in wide Josephson junctions that can also lead to a JDE, it is shown that such extrinsic effects can be suppressed by using a criss-crossed measurement geometry. The junctions are shown to behave as 'tunable second-order $\varphi_0$-junctions', in which the supercurrent transport can be tuned between Cooper pairs and Cooper quartets of charges $2e$ and $4e$ respectively, through the analysis of $\Delta I_c$ in the Fraunhofer interference pattern and comparison with the proposed $\delta$-dependent CPR. The simulated $\Delta I_c$ with a strong second harmonic term $\left(\frac{I_2}{I_1} \approx 0.4\right)$, as inferred from the CPR analysis well replicates the experimental behavior. $\Delta I_c$ is also shown to have nodes at every $\left(\frac{\Phi_0}{2}\right)$, further confirming the validity of the proposed $\delta$-dependent CPR. Besides being important for the observation of a JDE, this CPR has unique properties such as the controlling the relative phase difference between the two harmonics in the junction and controlling the relative direction of supercurrent flow by tuning the $(\delta, \varphi)$ phase space. Josephson junctions of topological materials have been explored largely in the context of topological superconductivity[74,75] and though the protection against backscattering offered by the topological states leading to higher order Andreev processes has been reported in junctions prior to this work[50,63,66], their role in the creation of a large JDE and controlling its magnitude has been unambiguously identified and explored in this work, making them more relevant in creating supercurrent diodes of much larger efficiencies. Moreover, they would also be useful in the study of the $4e$ Cooper quartet transport without the need for multiple superconducting terminals[78,79], which have also been predicted to be useful in the creation of parity protected superconducting qubits[80,81]. We would also like to note the recent observation of 4e supercurrents in superconducting quantum interference devices (SQUIDs) consisting of InAs-Al heterostructure[82].

## Methods

### Exfoliation

Thin flakes of PtTe$_2$ were exfoliated from a single crystal of 1T-PtTe$_2$ (purchased from HQ Graphene) under ambient conditions on a Si/SiO$_2$ substrate using a Nitto adhesive tape (SPV 224) and standard exfoliation techniques. Very thin flakes of few layer thicknesses are hard to obtain due to stronger interlayer attraction present in PtTe$_2$. A thin flake of around 17.5 nm thickness with a relatively large area was identified with the help of an optical microscope and its thickness was determined using an atomic force microscope (AFM). This flake was then used to fabricate the Josephson junctions presented in the main text.

### Device fabrication

The Josephson junctions were fabricated on this flake using electron-beam lithography. The substrate containing the flake was spin-coated at 4000rpm with a positive resist AR-P 669.04 and annealed at 150 $^\circ$C for 60 seconds followed by the same procedure for AR-P 679.03 (purchased from Allresist GmBH). The substrate was then exposed to the electron beam at 10 kV energy and developed using AR 600-56 for 90 seconds. After development and gentle ion milling to remove residual resist on top surface, superconducting electrodes Ti (2 nm)/Nb (40 nm)/Au (4 nm) substrate was sputtered on the substrate. The lift-off was performed by immersing the substrate in acetone overnight.

### Electrical measurements

Electrical measurements were performed in a Bluefors LD-400 dilution refrigerator with a bottom-loading probe and a base temperature of 20 mK. The fridge is equipped with RF and RC filters (from QDevil Aps) that help decrease the electron temperature during measurements. DC measurements were performed to obtain the current-voltage characteristics of the Josephson junctions. The current bias was applied through a Keithley 6221 current source and the voltage was measured using a Keithley 2182 A nanovoltmeter. A two-dimensional superconducting vector magnet attached to the system was used to control the magnetic field and measure the oscillations in the interference under different in-plane magnetic fields. The critical current is defined as the current at which the derivative of voltage with respect to the current exceeds a specific threshold.

## Data availability

The data that support the findings of this study are available from the corresponding authors upon reasonable request.

## Code availability

All relevant code used in this study is available from the corresponding authors upon reasonable request.

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

## Acknowledgements

P.K.S. would like to thank Niels Schröter for useful discussions. S.S.P.P. acknowledges support from the European Union (European Research Council Advanced Grant SUPERMINT, project number 101054860). Views and opinions expressed are however those of the authors only and do not necessarily reflect those of the European Union or the European Research Council. Neither the European Union nor the granting authority can be held responsible for them. M.T.A. acknowledges support from the NSF through the University of Illinois at Urbana-Champaign Materials Research Science and Engineering Center DMR-1720633.

## Author contributions

P.K.S. and S.S.P.P. conceived the project. P.K.S. performed the exfoliation and fabrication of lateral junctions. J.-K.K. fabricated the vertical Josephson junctions. Y.W. fabricated the Hall bar devices. P.K.S. performed all the electrical measurements with support from A.D. P.K.S. and M.T.A. performed all the data analysis with help from A.K.P. P.K.S. and M.T.A. performed the CPR analysis. M.T.A. performed all the self-consistent CPR simulations with assistance from G.J.C. and M.G. P.K.S., M.T.A., M.G. and S.S.P.P. discussed the data and wrote the manuscript with contributions from all authors.

## Funding

## Competing interests

The authors declare no competing interests.
