## [Transparent Peer Review file · Communications Physics]

Long-range Phase Coherence and Tunable Second Order ϕ_0 -Josephson Effect in a Dirac Semimetal 1T-PtTe₂

Corresponding Author: Professor Stuart Parkin, Mr Pranava Keerthi Sivakumar

Version 0:

Reviewer comments:

Reviewer #1

(Remarks to the Author)

The manuscript by Sivakumar et al. investigates the so-called intrinsic superconducting diode effect (SDE) in Josephson junctions based on the Dirac semimetal 1T-PtTe as a weak link. This work is a timely investigation of an intensely studied subject, which I have read with interest. Both the experiments and the data analysis are conducted with great care, in order to separate the interesting effects from the spurious ones. The quality and impact of this work is remarkable. I would recommend the publication in Communication Physics provided my questions below are answered (in particular the Major points).

Major points.

1) The key quantity for the interpretation of the data, according to the authors, is δ , which must be nonzero in order to observe a finite SDE. As far as I deduce from the manuscript, δ is nonzero based on the GL analysis in Ref.4. But, also in Ref 4, it is just *assumed* that such a quantity exists and is nonzero (at least, the discussion before Eq 6 of Ref 4 does not clarify the point, in my opinion). I think this point is not really settled in the literature. So far, there are instead simple arguments that would suggest that δ could be indeed zero. An illustration is provided by Yokoyama, Eto and Nazarov [Phys. Rev. B 89, 195407 (2014), see last section and Fig 10 therein]. In summary, according to the spin-orbit based mechanism (assumed also in this work) the anomalous ϕ_0 shift emerges from the difference in momentum between left and right movers (Andreev bound states in the weak link), multiplied by the length L of the junction. Now, higher harmonics can be seen as originating from higher orders of reflection for Andreev bound states. Since the n -th harmonic corresponds to a traveled distance nL , it seems then reasonable that ϕ_0 for the n th harmonics is n times the ϕ_0 of the 1st, i.e., that δ should be zero.

On the other hand, the authors here are not considering the fact that there are many channels in a junction of finite width (by the way, can the authors estimate how many?). Each channel would have a different Fermi velocity (and thus k_x) along x , and this affects ϕ_0 (in the short ballistic case described by Buzdin [Phys. Rev. Lett. 101, 107005 (2008)], ϕ_0 scales as $(v_F)^{-2}$). Now, the difference in ϕ_0 of, say, two channels both with skewed CPR (thus with a significant second harmonic) might mimic the effect of δ . Notice that the second harmonic is here still necessary, since the sum of sine functions is always a sine.

I think this point (the fact that δ can effectively be nonzero, the role of multiple transverse channels) is still not settled in the literature; several (especially theory) works assume a nonzero δ , but it is not clear to me why it should be effectively nonzero.

Arriving to my question: can the author justify a finite δ ? Or, even better, can the authors **use** their findings to support a finite δ hypothesis? Can the authors justify the fact that the many channels (with different k_x and presumably different ϕ_0) can be ignored in favor of a "single channel" picture?

2) I assume that the simulations in Fig 3 and 4 are based on Eq 4 (correct?). My question is: why **in the simulations** of Fig3 the ΔI_c does not cross zero at exactly half-flux (as mentioned in line 267 and deduced from Eq 4), but at higher B_z compared to the half-flux field? The authors provide physical arguments why in experiments the nodes are lifted, but in simulations the nodes are there and they should be at half-flux, but in Fig 3m,n,o,p they are not. Why? As a side remark, in the main text of Ref.10 precisely these nodes at half flux are predicted, measured and discussed (see extended data Fig2d). In that case the nodes are exactly at half flux.

3) Whenever a SDE is induced by an in-plane field perpendicular to the current, one should be careful that at least another alternative mechanism exists that might generate the effect. This is the mechanism found by Davydova et al. [M. Davydova, S. Prembabu, and L. Fu, *Sci. Adv.* 8, eabo0309 (2022).] and discussed further by Banerjee et al. [Phys. Rev. Lett. 131, 196301 (2023)]. In this case, Meissner currents in the Nb leads, coupled to the PtTe₂, induced in this latter a finite momentum superconductivity. As a result, ϕ_0 and SDE are induced, which are hard to distinguish from the Rashba based mechanism. Can the authors at least comment on this possibility?

Minor points (indeed the point 9 is not so minor since it affects the title).

4) In fig1b, the axis z is not clear: it points at 45 degrees, indicating in 3D that it is out of plane, but it is not immediate to understand whether out-of-substrate or into-the-substrate. Can this be made clearer?

5) Fig2c and line 138. It is not so clear (the error bars are too large) that the tail of ΔI_c near T_j is $\sim(T-T_j)^2$. This statement is too strong given the experimental uncertainty, in my opinion.

6) Line 105: how do you compute it and what is the value of λ_j in your case?

7) Line 109. "It is shown that in such junctions...". Where is it shown? Please, provide a reference.

8) Lines 177-181. The way the text is written is not clear. If ϕ_1 and ϕ_2 are both zero, this is a skewed junction without ϕ_0 ($I=0$ at $\phi=0$). "Furthermore setting $I_2=0$ " (line 180) would produce a simple sine CPR without ϕ_0 . I think it is just a matter of rephrasing in a more clear fashion what the authors mean.

9) Actually I am not sure if this should be a major point, instead. The authors call this phenomenology "second order ϕ_0 -junction". But, in my view, these are two distinct features (having a second harmonic in the CPR and having an anomalous shift in the CPR), which in some cases can appear together. According to the authors, this should not even be sufficient to see the effect, because if one has one channel with second harmonic and $\phi_2=2\phi_1$ (i.e. $\delta=0$), then no SDE is observed. This point would be in principle "minor", but the authors themselves put this expression in the title, therefore I would like to read a convincing argument to call this phenomenon in this way (and why it should go in the title).

That said, I would anyway in line 187 add after "...as that in Equation (3)..." the words "with nonzero δ ". I would also change "is dubbed" in "We shall call ..." (changing the sentence in the active form).

10) Lines 252-265. There is a repetition: the authors probably wrote the same paragraph twice in slightly different forms and possibly forgot to delete the old version. Please rewrite the paragraph eliminating the repeating sentences.

Reviewer #2

(Remarks to the Author)

Sivakumar et al. investigate Josephson junctions fabricated on thin flakes of type-II Dirac semimetal candidate 1T-PtTe₂. The authors observe the Josephson diode effect, manifesting as an asymmetry of the critical supercurrent in positive versus reverse bias that turns on in the presence of "small" magnetic field B_y , applied through the cross-section of the device, perpendicular to the direction of supercurrent flow. The effect turns on sharply at about $B_y=10$ mT, remains constant until about $B_y=50$ mT, before sharply dropping to zero again at about $B_y=70$ mT. The effect goes away when the in-plane field is parallel to the direction of supercurrent direction.

The authors stress that they do not observe any diode effect when a perpendicular magnetic field is applied through the sample, only a standard Fraunhofer effect is observed. However, the Fraunhofer effect acquires a critical current asymmetry when an additional B_y is turned on. This is modeled as a phase shift δ between the first and second harmonic terms of the current-phase relationship (CPR). Particularly, the authors suggest that the critical current asymmetry vanishes every half-flux quantum. This is putatively a major highlight of the work.

Although several aspects of this work appear in previous papers by the same group, I find certain aspects of this work interesting that go beyond what has been previously attempted. First, a direct measurement of the $\phi_0/2$ effect in the diode efficiency has not been clearly observed before. Although I am not fully convinced by the authors' interpretation yet (comments below), I think the approach is quite interesting, and perhaps helps distinguish from the more trivial mechanisms for the superconducting diode effect that have been discovered over the last few years. Second, I think that large critical current Josephson junctions that show pronounced second-harmonic content in the CPR are interesting and useful to the community. Finally, I think it is interesting that the 2D material systems such as PtTe₂ are slowly approaching superconductor-semiconductor hybrid materials in terms of device cleanliness and performance, already with mean-free paths that are hundreds of nanometers and junction transparencies of order 0.5. Although more progress will be needed, band geometry and topological effects in these materials are strong, and the surfaces are pristine owing to the 2D nature. This can be important for technological applications. However, I have several comments and criticisms that the authors should try to address before I can recommend publication.

1. What factors determine the field scale of about $B_y \sim 50$ mT that the authors need to observe the diode effect?

2. Relating to point (1), how does this field scale compare with the in-plane cross-section of the device? The "oscillatory" structure of the critical current and diode efficiency (Fig. 2a, b) looks like an in-plane Fraunhofer effect, with a field scale of about 70 mT, which roughly matches the field scale where the diode effect is observed. Can the authors comment?

3. The authors mention that the helicity of spin-momentum locking alternates between the chalcogen layers across the

thickness of the sample. Does this change the diode-effect mechanism considered for example in Ref. 11? Does the diode efficiency increase or decrease compared with regular Rashba spin-orbit coupled electronic states?

4. The authors claim, rather strongly, that three effects point to two-dimensional helical spin-momentum locking of electrons in their Josephson junctions: (i) The T^2 dependence of the diode asymmetry, (ii) the absence of a vertical diode effect (iii) The disappearance of the diode effect when the in-plane field is applied along the direction of the supercurrent. About (i), I think the T^2 dependence is very hard to see in Fig. 2(c), the data looks more or less linear, and the T^2 fitting in the inset has only four points. While point (ii) is an interesting experiment, it is not clear to me why it supports an in-plane helical spin-locking mechanism, rather than only suggesting that a vertical symmetry breaking is *not* taking place. Finally, (iii) may not need any spin-momentum locking at all and may originate from the orbital effect of the in-plane magnetic field (Sundaresh et al, Nature Communications volume 14, Article number: 1628 (2023), Banerjee et al, Physical Review Letters 131.19 (2023) 196301, Davydova et al, Science advances 8.23 (2022): eabo0309, Nakamura et al, PRB 109.9 (2024): 094501.). Perhaps some additional discussion is warranted.

5. Regarding the above point, I think the authors should make a softer claim as far as spin-momentum locking of the electronic states is concerned, especially since a purely orbital-effect origin of the diode effect has been proposed (Davydova et al, Science Advances 8.23 (2022): eabo0309, PRB 109.9 (2024): 094501) and observed in experiments (Sundaresh et al, Nature Communications volume 14, Article number: 1628 (2023), Banerjee et al, PRL 131.19 (2023),196301). If the authors can rule out these effects by further analysis/arguments, that would be fine too.

6. Next, the authors study the Fraunhofer effect of their Josephson junctions, in the presence of a vertical magnetic field (B_z), while simultaneously applying an in-plane magnetic field B_y . The authors observe an interesting asymmetry of the Fraunhofer pattern in the presence of a finite B_y , even when they have corrected for magnetic field misalignment (Fig. S6). While the authors have relegated this experiment to the supplement, I think it is important. This to me is a tell-tale signature of the in-plane magnetic field creating a Fraunhofer-like current distribution across the cross-section of the Josephson junction. Did the authors try reversing the direction of B_y ? Does that change the sign of this asymmetry? Similar effects have been observed in InAs-Al Josephson junctions: (Ref. 21, PRB 95, 035307 (2017), Rasmussen et al. PRB B 93.15, 155406(2016)). The authors mention flux-focussing discussed in Ref. 21 but do not discuss the in-plane Fraunhofer effect in much detail. I think it is important to discuss this effect. More importantly, the authors should comment on whether this effect (similar in spirit to the orbital effect I mentioned above) can influence their diode-effect signatures.

7. By performing a Fraunhofer analysis of the critical current asymmetry with increasing B_y , the authors observe an interesting structure of ΔI_c with out-of-plane flux. This structure changes with increasing B_y , which is modeled as a phase difference between the first and second harmonics of the current phase relation. Although the theory and experiment seem to qualitatively agree, quantitatively I don't see a strong match. Particularly, the $\Phi_0/2$ disappearance of ΔI_c , although clearly observed in theory is very hard to see in the experimental data. Given that the authors claim a "Second order Φ_0 Josephson effect", I think more convincing analysis or data is needed. In the very least, the authors should clearly point out the $\Phi_0/2$ features that the readers are meant to observe in the data.

8. In fact, this discrepancy becomes clearer by comparing Figs. 4(f) and 4(g) where I_{c+} is compared with the first Fraunhofer lobe for theory and experiment. Although the "dip" at zero flux is captured very well by theory, two peaks of I_{c+} occur at roughly $\Phi = \Phi_0/2$ in theory, but $\Phi = \Phi_0$ in the experiment. Can the authors comment?

9. The authors mention large ratios of g/vF in the Discussion section but do not provide any quantitative estimate. Can the authors estimate this factor (even within an order of magnitude) using their experimental values of ΔI_c , for instance?

10. I think a schematic showing the spin-momentum locking structure on the Fermi surface of the material would be helpful to the reader, especially since the authors seem to suggest that the Fermi surface does not display usual Rashba-type spin-momentum locking. I also wonder whether the more complicated electronic structure plays any role in the "protection from backscattering" that the authors indicate in the schematic of Fig. 5(g). For example, if an electronic channel with opposite spin-momentum helicity is also available at the Fermi surface (which seems to be the case?), the backscattering protection mechanism should break down.

Version 1:

Reviewer comments:

Reviewer #1

(Remarks to the Author)

The authors have answered the Referees' questions in an appropriate fashion. I confirm my positive judgement and I recommend publication in "Communications physics".

Reviewer #2

(Remarks to the Author)

I thank the authors for their thoughtful and detailed responses. The authors have addressed more or less all my concerns.

However, one issue still bothers me. The g/vF ratio estimated by the authors is $\approx 10^{-3}$ m/s. Given that $vF \approx 3.3 \cdot 10^5$ m/s in this material, we get $g = 330$! This is an astoundingly large g -factor, much larger than the $g \approx 50$ that the authors cite in Ref. 58. The authors should mention this clearly in the manuscript and provide a suitable justification for the readers. It is important because this is central to the spin-orbit interpretation of the diode effect. I am happy to recommend publication with this change included.

Version 2:

Reviewer comments:

Reviewer #2

(Remarks to the Author)

The authors have provided a reasonable justification for the relatively large g -factors implied by their experiments. I appreciate their detailed response. I confirm my recommendation for publication.

Reviewers' comments:

Reviewer #1 (Remarks to the Author):

The manuscript by Sivakumar et al. investigates the so-called intrinsic superconducting diode effect (SDE) in Josephson junctions based on the Dirac semimetal 1T-PtTe as a weak link. This work is a timely investigation of an intensely studied subject, which I have read with interest. Both the experiments and the data analysis are conducted with great care, in order to separate the interesting effects from the spurious ones. The quality and impact of this work is remarkable. I would recommend the publication in Communication Physics provided my questions below are answered (in particular the Major points).

We thank the reviewer for their interest and favorable evaluation and recognition of our meticulous work. We hope to satisfactorily address all the questions in their report.

Major points.

1) The key quantity for the interpretation of the data, according to the authors, is δ , which must be nonzero in order to observe a finite SDE. As far as I deduce from the manuscript, δ is nonzero based on the GL analysis in Ref.4. But, also in Ref 4, it is just *assumed* that such a quantity exists and is nonzero (at least, the discussion before Eq 6 of Ref 4 does not clarify the point, in my opinion). I think this point is not really settled in the literature. So far, there are instead simple arguments that would suggest that δ could be indeed zero. An illustration is provided by Yokoyama, Eto and Nazarov [Phys. Rev. B 89, 195407 (2014), see last section and Fig 10 therein]. In summary, according to the spin-orbit based mechanism (assumed also in this work) the anomalous ϕ_0 shift emerges from the difference in momentum between left and right movers (Andreev bound states in the weak link), multiplied by the length L of the junction. Now, higher harmonics can be seen as originating from higher orders of reflection for Andreev bound states. Since the n -th harmonic corresponds to a traveled distance nL , it seems then reasonable that ϕ_0 for the n th harmonics is n times the ϕ_0 of the 1st, i.e., that δ should be zero.

On the other hand, the authors here are not considering the fact that there are many channels in a junction of finite width (by the way, can the authors estimate how many?). Each channel would have a different Fermi velocity (and thus k_x) along x , and this affects ϕ_0 (in the short ballistic case described by Buzdin [Phys. Rev. Lett. 101, 107005 (2008)], ϕ_0 scales as $(v_F)^{-2}$). Now, the difference in ϕ_0 of, say, two channels both with skewed CPR (thus with a significant second harmonic) might mimic the effect of δ . Notice that the second harmonic is here still necessary, since the sum of sine functions is always a sine.

I think this point (the fact that δ can effectively be nonzero, the role of multiple transverse channels) is still not settled in the literature; several (especially theory) works assume a nonzero δ , but it is not clear to me why it should be effectively nonzero.

Arriving to my question: can the author justify a finite δ ? Or, even better, can the authors **use** their findings to support a finite δ hypothesis? Can the authors justify the fact that the many channels (with different k_x and presumably different ϕ_0) can be ignored in favor of a "single channel" picture?

We thank the referee for their insightful and meticulous feedback. We concur with the referee's assessment that in a single channel approximation, a phase shift ϕ is induced. Specifically, for a ballistic channel [PRL 101, 107005 (2008)], ϕ scales as $\phi \propto \frac{\alpha L B_y}{v_F}$, where α denotes the strength of the spin-orbit coupling (SOC),

L is the junction length, v_F is the Fermi velocity. We attribute our CPR's characteristics to two key factors: strong second-order harmonics and the presence of multiple transport channels. The higher harmonics, highlighted by the referee, likely stem from higher orders of reflection of Andreev bound states, such that $\phi_2 = 2\phi_1$. Comparing our junction's actual length of 390 nm to the superconducting coherence length $\xi_0 \approx 200$ nm suggests that we are not in either the long or short junction limit. Additionally, at $T = 20$ mK, the

thermal coherence length in the junction $\xi_T = \frac{\hbar v_F E}{\pi \Delta} \approx 14 \mu\text{m}$, is significantly larger than both ξ_0 and junction length. With a junction width $W \approx 5 \mu\text{m}$, numerous Andreev bound states (ABSs) with nonzero k_y are likely present, facilitating supercurrent transport. This suggests the existence of a large number of channels with ABS energies characteristic of a long junction regime. In an ideal clean long SNS junction, ABS energies disperse linearly with the phase difference, resulting in a sawtooth-shaped CPR at low temperatures [Rev. Mod. Phys. 76, 411 (2004)]. Alternatively, including disorder leads to a smoother CPR [PRL 123, 026802 (2019)], primarily featuring contributions from the first and second harmonics. Therefore, our junction benefits from a relatively clean interface, wide junction, and a significant number of ABSs, contributing prominently to the strong second-order harmonic observed in our CPR. As a result, a CPR with N transport channels can be written as follows:

$$I = \sum_{i=1}^N I_{1i} \sin(\phi + \phi_i) + I_{2i} \sin(2\phi + 2\phi_i) = I_1 \sin(\phi + \varphi_1) + I_2 \sin(2\phi + \varphi_2),$$

where

$$\varphi_1 = \tan^{-1} \frac{\sum_{i=1}^N I_{1i} \sin \phi_i}{\sum_{i=1}^N I_{1i} \cos \phi_i}, \quad \varphi_2 = \tan^{-1} \frac{\sum_{i=1}^N I_{2i} \sin 2\phi_i}{\sum_{i=1}^N I_{2i} \cos 2\phi_i}.$$

From this expression, we define $\delta = \varphi_2 - 2\varphi_1$, which generally need not be equal to zero. To get an estimate for N , we consider a finite width W to quantize the transverse wave vectors, $k_{yi} \approx i \pi / W$, where $W \approx 5 \mu\text{m}$, and consider a linear energy dispersion near the Fermi level $E_F \approx \hbar v_F k_F$. As a result, the number of propagating modes at the Fermi level is $N \approx \frac{E_F}{\pi \hbar v_F} \approx E_F (eV) \times 10^3$ which comes out

order of thousands.

2) I assume that the simulations in Fig 3 and 4 are based on Eq 4 (correct?). My question is: why ****in the simulations**** of Fig3 the ΔI_c does not cross zero at exactly half-flux (as mentioned in line 267 and deduced from Eq 4), but at higher B_z compared to the half-flux field? The authors provide physical arguments why in experiments the nodes are lifted, but in simulations the nodes are there and they should be at half-flux, but in Fig 3m,n,o,p they are not. Why? As a side remark, in the main text of Ref.10 precisely these nodes at half flux are predicted, measured and discussed (see extended data Fig2d). In that case the nodes are exactly at half flux.

The simulations in Fig.3 (m-p) show that the nodes in ΔI_c at higher B_z happen exactly at half flux quantum as predicted by equation (4) but we also show that the experimental occurrence of lifted nodes at low B_z can also captured in our simulations by the introduction of an additional phase difference $\left(\beta \frac{I_{tot}}{I_1} \frac{1}{y W} \right)$, as we are in the wide junction limit. This is explained in lines 241-250 of the main text. Without this term, all nodes

in the JDE occur at half flux quantum. Below we show simulated ΔC as a function of Φ/Φ_0 with $\beta = 0$, $I_2 = 0.4$, $\delta = \pi/2$,
 I_1

We also note that Eq 4 is obtained under the assumption of $\alpha_{I2} \ll 1$. For intermediate values of α_{I2} , Eq 4 is only a good approximation and the self-consistent numerical simulations, may slightly deviate from Eq 4.

The InAs junctions used in Ref[10] with much lower critical currents are most likely still in the short (narrow) junction limit where self-field effects are less significant and hence the first node is clearly visible unlike in our case.

3) Whenever a SDE is induced by an in-plane field perpendicular to the current, one should be careful that at least another alternative mechanism exists that might generate the effect. This is the mechanism found by Davydova et al. [M. Davydova, S. Prembabu, and L. Fu, *Sci. Adv.* 8, eabo0309 (2022).] and discussed further by Banerjee et al. [*Phys. Rev. Lett.* 131, 196301 (2023)]. In this case, Meissner currents in the Nb leads, coupled to the PtTe₂, induced in this latter a finite momentum superconductivity. As a result, ϕ_0 and SDE are induced, which are hard to distinguish from the Rashba based mechanism. Can the authors at least comment on this possibility?

We do agree with the reviewer and the other reports that finite momentum in Josephson junctions can also be induced by orbital effects and is hard to disentangle completely from finite momentum Cooper pairs generated through spin-orbit coupling. We believe that there is possibly a synergy of both these mechanisms that is needed to generate a significantly diode effect. In our case, we believe the spin-orbit coupling mechanism dominates due to the following reasons. We have included some of these arguments also in the manuscript.

- The observation of a diode effect due to orbital effects in the superconductor is well described for short, ballistic junctions [M. Davydova, S. Prembabu, and L. Fu, *Sci. Adv.* 8, eabo0309 (2022)] which is also the picture used to explain experimental observations [Banerjee et al. *Phys. Rev. Lett.* 131, 196301 (2023)], where the junctions are well within the ballistic regime. This leads to perfect transparency in these junctions ($\tau = 1$) and allows the transport of finite momentum Cooper pairs generated by orbital effects. As the junction transparency decreases, the Josephson diode effect generated by this mechanism also drops significantly as shown in Fig.3B of *Sci. Adv.* 8, eabo0309 (2022) and in Fig.2 of *Phys. Rev. B* 109, 024504 (2024). In our case, the junction separations are in the quasi-diffusive regime and have transparencies (τ) around 0.45 (Refer SI section 12), in which case the contribution of this mechanism to the observed diode effect would theoretically have maximum efficiency below 4% according to these works. The experimentally observed efficiency in our case is close to 32% that is 8 times higher and cannot be accounted for completely by finite momentum Cooper pairs created by orbital effects only.

- The thickness of the niobium electrodes (40nm) used in our experiments are much smaller than the London penetration depth of niobium (120nm) [Ref. Phys. Rev. B **72**, 064503 (2005)] for the in-plane magnetic fields to create significant Meissner currents.
- It is also to be noted that most, if not all of the experimental studies that report diode effect due to orbital effects are also in Josephson junctions made of materials with high spin-orbit coupling materials like InAs. In order to extract contributions of the orbital effect, the experiments need to be performed in a similar geometry with low spin-orbit coupling materials.

Minor points (indeed the point 9 is not so minor since it affects the title).

4) In fig1b, the axis z is not clear: it points at 45 degrees, indicating in 3D that it is out of plane, but it is not immediate to understand whether out-of-substrate or into-the-substrate. Can this be made clearer?

We have modified this figure to make it clearer, as suggested.

5) Fig2c and line 138. It is not so clear (the error bars are too large) that the tail of Delta Ic near Tj is $\sim(T - T_j)^2$. This statement is too strong given the experimental uncertainty, in my opinion.

We have remeasured the critical currents in the same device (after 2 years) due to its robust air stability with more data points and better precision this time and have updated the graph and fit to support our hypothesis. These results are now included in the main manuscript.

This figure shows the temperature dependence of the diode effect measured in device B1 at $B_{\parallel} = 24mT$. The fit represents a $(T - T_j)^2$ dependence with $T_j \approx 1.4K$. Inset shows the measurement of resistance vs temperature in the presence of a magnetic field $B_{\parallel} = 24mT$, which shows T_j to be around $1.4K$.

6) Line 105: how do you compute it and what is the value of λ_j in your case? The calculation and the estimation of A_i in terms of the junction width are provided in the SI section 3.

We define a constant parameter $a = \phi_0 / (2n\mu_0 d_{\text{eff}} J_c A_i^2)$, which is a dimensionless coefficient determining the influence of the out-of-plane magnetic field on the phase gradient induction in the junction. By setting $B_y = 0$ and applying same-side and opposite-side biasing in our simulations, we

determine $a := 1/2$.
1. This corresponds to the well-known expression $A_i = \left(\frac{43e}{2n\mu_0 d_{\text{eff}}} \right)$. Our estimates indicate $A_i := 2w = 3\mu\text{m}$ for device B1.

7) Line 109. "It is shown that in such junctions...". Where is it shown? Please, provide a reference. Reference to the supplementary information has been added now.

8) Lines 177-181. The way the text is written is not clear. If ϕ_1 and ϕ_2 are both zero, this is a skewed junction without ϕ_0 ($I=0$ at $\phi=0$). "Furthermore setting $I_2=0$ " (line 180) would produce a simple sine CPR without ϕ_0 . I think it is just a matter of rephrasing in a more clear fashion what the authors mean.

The lines have been rephrased to make it clearer now.

9) Actually I am not sure if this should be a major point, instead. The authors call this phenomenology "second order ϕ_0 -junction". But, in my view, these are two distinct features (having a second harmonic in the CPR and having an anomalous shift in the CPR), which in some cases can appear together. According to the authors, this should not even be sufficient to see the effect, because if one has one channel with second harmonic and $\phi_2=2\phi_1$ (i.e. $\delta=0$), then no SDE is observed. This point would be in principle "minor", but the authors themselves put this expression in the title, therefore I would like to read a convincing argument to call this phenomenon in this way (and why it should go in the title). That said, I would anyway in line 187 add after "...as that in Equation (3)..." the words "with nonzero δ ". I would also change "is dubbed" in "We shall call ..." (changing the sentence in the active form).

PtTe₂ is intrinsically a two dimensional material and exfoliates as thick and wide flakes usually more than 1 \$\mu\text{m}\$ in width and > 10nm in thickness and as we have argued in an earlier point, there are multiple channels (1000s) present in our devices in this limit. Our analysis is certainly valid in two or more dimensions, where there are multiple channels present. The scenario discussed by the reviewer is an extreme case of reduced dimensionality (~nm width, junction separation less than 100nm and less than 5 layer thick) that certainly would affect the band structure and material properties of PtTe₂, as is the case for most materials.

The flakes are exfoliated in ambient conditions and the interface of PtTe₂ has been exposed to air before depositing the superconductor on top. One might assume that the junction interfaces are not as pristine as devices fabricated with superconductor interfaces in-situ yet we observe such a strong second harmonic supercurrent in these junctions, even in the presence of only moderate transparency. Hence, it can be argued that the appearance of a second harmonic component is quite 'intrinsic' to Josephson junctions of PtTe₂ rather than being strongly dependent on the quality of the interface, as in certain semiconductor junctions, where a pristine interface quality is a necessity to observe such effects and it can drastically affect the properties of the junction. We expect any Josephson junction fabricated under ambient conditions from a PtTe₂ flake or crystal and is not in the single channel limit to display the observed properties. Hence, the use of this name, we believe, is quite justified in our case.

The tunability of the anomalous phase shift between the two harmonics is also an intrinsic property of PtTe₂ as we have demonstrated with ample evidence in this work. The main point to be noted here is that the

anomalous phase shift between the harmonics is ‘tunable’ in our case from zero to create a non-zero delta (and a JDE) through an in-plane magnetic field and this tunability of delta in our work is clearly demonstrated through various measurements of the diode effect. This is why we have the term ‘tunable’ in our title. We will also include this term to all references of the CPR to make this point clearer.

10) Lines 252-265. There is a repetition: the authors probably wrote the same paragraph twice in slightly different forms and possibly forgot to delete the old version. Please rewrite the paragraph eliminating the repeating sentences.

We apologize for this mistake and have corrected it.

Reviewer #2 (Remarks to the Author):

Sivakumar et al. investigate Josephson junctions fabricated on thin flakes of type-II Dirac semimetal candidate 1T-PtTe₂. The authors observe the Josephson diode effect, manifesting as an asymmetry of the critical supercurrent in positive versus reverse bias that turns on in the presence of "small" magnetic field B_y , applied through the cross-section of the device, perpendicular to the direction of supercurrent flow. The effect turns on sharply at about $B_y=10$ mT, remains constant until about $B_y=50$ mT, before sharply dropping to zero again at about $B_y=70$ mT. The effect goes away when the in-plane field is parallel to the direction of supercurrent direction.

The authors stress that they do not observe any diode effect when a perpendicular magnetic field is applied through the sample, only a standard Fraunhofer effect is observed. However, the Fraunhofer effect acquires a critical current asymmetry when an additional B_y is turned on. This is modeled as a phase shift δ between the first and second harmonic terms of the current-phase relationship (CPR). Particularly, the authors suggest that the critical current asymmetry vanishes every half-flux quantum. This is putatively a major highlight of the work.

Although several aspects of this work appear in previous papers by the same group, I find certain aspects of this work interesting that go beyond what has been previously attempted. First, a direct measurement of the $\Phi_0/2$ effect in the diode efficiency has not been clearly observed before. Although I am not fully convinced by the authors' interpretation yet (comments below), I think the approach is quite interesting, and perhaps helps distinguish from the more trivial mechanisms for the superconducting diode effect that have been discovered over the last few years. Second, I think that large critical current Josephson junctions that show pronounced second-harmonic content in the CPR are interesting and useful to the community. Finally, I think it is interesting that the 2D material systems such as PtTe₂ are slowly approaching superconductor-semiconductor hybrid materials in terms of device cleanliness and performance, already with mean-free paths that are hundreds of nanometers and junction transparencies of order 0.5. Although more progress will be needed, band geometry and topological effects in these materials are strong, and the surfaces are pristine owing to the 2D nature. This can be important for technological applications. However, I have several comments and criticisms that the authors should try to address before I can recommend publication.

We thank the reviewer for their thorough note on the interesting aspects of our work and hope to answer all questions and comments satisfactorily.

1. What factors determine the field scale of about $B_y \sim 50$ mT that the authors need to observe the diode effect?

The diode effect begins to appear just as soon as B_y is increased from zero and does not need the application of a 50 mT field. If the reviewer is referring to magnetic field at which the diode effect is maximized, it is directly related to the finite momentum of the Cooper pairs induced by the Zeeman effect and the junction length, as has been studied in detail in previous works. Please refer *Nat. Phys.* **13**, 87–93 (2017); *Nat. Commun.* **9**, 3478 (2018) and *Nat. Phys.* **18**, 1228–1233 (2022) for more details.

The maxima happens when the phase shift $S = 2q \cdot d$ (where $2q$ is the additional momentum and d is the effective separation between the two electrodes) is equal to π_2 .

The finite momentum is proportional to the Zeeman energy and is given by: $2q = \frac{2g\mu_B \gamma}{h v_F}$ where g is the gyromagnetic ratio, i is the Bohr magneton, h is the reduced Planck's constant and v_F is the Fermi velocity.

These are the parameters that determine the magnetic field at which the diode effect is maximized. In our case, we observe the diode effect to maximize around $B_y = 24$ mT and decreases thereafter. In principle, the diode effect can be observed as long as supercurrents flow through the junction but it becomes vanishingly small to extract directly from the Fraunhofer pattern at larger values of B_y or smaller critical currents.

2. Relating to point (1), how does this field scale compare with the in-plane cross-section of the device? The "oscillatory" structure of the critical current and diode efficiency (Fig. 2a, b) looks like an in-plane Fraunhofer effect, with a field scale of about 70 mT, which roughly matches the field scale where the diode effect is observed. Can the authors comment?

We note (as in the manuscript) that the application of B_y directly to the sample also creates an additional flux in the sample that distorts the observation of the diode effect which could either be due to the finite thickness of the sample or due to the ever-so slight misalignment of the magnetic field that induces an out-of-plane magnetic flux.

Addressing out-of-plane flux due to B_y :

This is the reason we use the Fraunhofer pattern to locate the exact critical current maxima where the net magnetic flux is zero and extract the value of the diode effect. This is clearly visible from the difference in the diode effect measured directly from an in-plane field as shown in Fig. 2b and the diode effect extracted from the effect of B_y only from the Fraunhofer pattern (by flux cancellation) as shown in Fig. 4c. The JDE shown in Fig. 4c is purely due to the effect of the in-plane field after removing the stray out-of-plane magnetic flux induced effects like Fraunhofer interference, as the net out-of-plane flux through the sample is zero. Hence, we recommend this method to avoid any pitfalls in studying the diode effect.

Addressing in-plane flux due to B_y :

The effective in-plane cross sectional area of the junction including the London penetration depth of the two superconducting electrodes is around $1.1725 \times 10^{-14} \text{ m}^2$, which gives the effective magnetic field needed to induce a single magnetic flux quantum in the junction is around 176 mT and the in-plane fields that we use in our study is much lower than this ($0 - 50$ mT).

Moreover, the evolution of I_c with S is quite different from what is expected for a typical flux-induced Fraunhofer pattern. There is an inherent asymmetry between I_c^+ and I_c^- that arises due to S as shown by the simulations in Fig. 4(b) and observed experimentally in Fig. 4(a). Rather than having nodes, the evolution of I_c with S has oscillations that decay slowly with no nodes in the critical current. Hence, we can also conclude that flux-induced Fraunhofer interference effects due to B_y in our junctions are not relevant to the observed diode effect.

We have also added these points to the manuscript to make our analysis/strategy clear to the reader.

3. The authors mention that the helicity of spin-momentum locking alternates between the chalcogen layers across the thickness of the sample. Does this change the diode-effect mechanism considered for example in Ref. 11? Does the diode efficiency increase or decrease compared with regular Rashba spin-orbit coupled electronic states?

The mechanism of the diode effect in this system is still the finite momentum Cooper pairing due to helical spin-momentum locking as discussed in Ref.11 and experimentally studied in Ref.4. The argument presented in the paper relates of the structure of these materials to the origin of the surface states with helical spin-momentum locking in these class of materials with 1T structure as has been studied in detail previously as given by Ref. 13. This provides a general avenue to look for Van der Waals materials with strong helical spin-momentum locking and explore supercurrent diode effect in materials with similar structure. The diode efficiency would still depend on the strength of spin-orbit coupling, which is material dependent.

4. The authors claim, rather strongly, that three effects point to two-dimensional helical spin-momentum locking of electrons in their Josephson junctions: (i) The T^2 dependence of the diode asymmetry, (ii) the absence of a vertical diode effect (iii) The disappearance of the diode effect when the in-plane field is applied along the direction of the supercurrent. About (i), I think the T^2 dependence is very hard to see in Fig. 2(c), the data looks more or less linear, and the T^2 fitting in the inset has only four points. While point (ii) is an interesting experiment, it is not clear to me why it supports an in-plane helical spin-locking mechanism, rather than only suggesting that a vertical symmetry breaking is *not* taking place. Finally, (iii) may not need any spin-momentum locking at all and may originate from the orbital effect of the in-plane magnetic field (Sundaresh et al, Nature Communications volume 14, Article number: 1628 (2023), Banerjee et al, Physical Review Letters 131.19 (2023) 196301, Davydova et al, Science advances 8.23 (2022): eabo0309, Nakamura et al, PRB 109.9 (2024): 094501.). Perhaps some additional discussion is warranted.

- We do not claim in the manuscript that the absence of diode effect in the vertical direction as an evidence for helical spin-momentum locking but rather it confirms that the polar axis is perfectly vertical, which results in the two-dimensional in-plane spin-momentum locking. It is as the reviewer states, an interesting additional experiment that demonstrates the absence of such an effect in the vertical direction and that the spin-momentum locking is essentially two-dimensional. The exact words used in the text are:
“Thus, this result shows the absence of net spin-momentum locking or any other finite momentum pairing mechanism when the current flows along the c-axis of PtTe₂ and, together with the results on LI, point to the existence of a two-dimensional helical spin-momentum locking in PtTe₂.”
However, we have rephrased it to avoid any confusion.
“Thus, this result shows the absence of net spin-momentum locking or any other finite momentum pairing mechanism when the current flows along the c-axis of PtTe₂. All the above results together point to the existence of a two-dimensional helical spin-momentum locking in PtTe₂.”
- We have remeasured the temperature dependence of the critical currents in the same device (after 2 years) due to its robust air stability with more data points and better precision this time and have updated the graph and fit to support our hypothesis. These results are now included in the main manuscript.

This figure shows the temperature dependence of the diode effect measured in device B1 at higher temperatures and $B_y = 24mT$. The fit represents a quadratic $(T - T_f)^2$ dependence with $T_f \approx 1.4K$. Inset shows the measurement of resistance vs temperature in the presence of a magnetic field $B_y = 24mT$, which shows T_f to be around $1.4K$

- With regard to the diode effect generated due to orbital effects, we will make this clearer in the manuscript. The answer to this comment is discussed in the next point, which is also similar.

5. Regarding the above point, I think the authors should make a softer claim as far as spin-momentum locking of the electronic states is concerned, especially since a purely orbital-effect origin of the diode effect has been proposed (Davydova et al, Science Advances 8.23 (2022): eabo0309, PRB 109.9 (2024): 094501) and observed in experiments (Sundaresh et al, Nature Communications volume 14, Article number: 1628 (2023), Banerjee et al, PRL 131.19 (2023),196301). If the authors can rule out these effects by further analysis/arguments, that would be fine too.

We do agree with the reviewer and the other reports that finite momentum in Josephson junctions can also be induced by orbital effects and is hard to disentangle completely from finite momentum Cooper pairs generated through spin-orbit coupling. We believe that there is possibly a synergy of both these mechanisms that is needed to generate a significantly diode effect. In our case, we believe the spin-orbit coupling mechanism dominates due to the following reasons. We have included some of these arguments also in the manuscript.

- The observation of a diode effect due to orbital effects in the superconductor is well described for short, ballistic junctions [M. Davydova, S. Prembabu, and L. Fu, Sci. Adv. 8, eabo0309 (2022)] which is also the picture used to explain experimental observations [Banerjee et al. Phys. Rev. Lett. 131, 196301 (2023)], where the junctions are well within the ballistic regime. This leads to perfect

transparency in these junctions ($t = 1$) and allows the transport of finite momentum Cooper pairs generated by orbital effects. As the junction transparency decreases, the Josephson diode effect generated by this mechanism also drops significantly as shown in Fig.3B of Sci. Adv. 8, eabo0309 (2022) and in Fig.2 of Phys. Rev. B 109, 024504 (2024). In our case, the junction separations are in the quasi-diffusive regime and have transparencies (t) around 0.45 (Refer SI section 12), in which case the contribution of this mechanism to the observed diode effect would theoretically have maximum efficiency below 4% according to these works. The experimentally observed efficiency in our case is close to 32% that is 8 times higher and cannot be accounted for completely by finite momentum Cooper pairs created by orbital effects only.

- The thickness of the niobium electrodes (40nm) used in our experiments are much smaller than the London penetration depth of niobium (120nm) [Ref. Phys. Rev. B **72**, 064503 (2005)] for the in-plane magnetic fields to create significant Meissner currents.
- It is also to be noted that most, if not all of the experimental studies that report diode effect due to orbital effects are also in Josephson junctions made of materials with high spin-orbit coupling materials like InAs. In order to extract contributions of the orbital effect, the experiments need to be performed in a similar geometry with low spin-orbit coupling materials.

6. Next, the authors study the Fraunhofer effect of their Josephson junctions, in the presence of a vertical magnetic field (B_z), while simultaneously applying an in-plane magnetic field B_y . The authors observe an interesting asymmetry of the Fraunhofer pattern in the presence of a finite B_y , even when they have corrected for magnetic field misalignment (Fig. S6). While the authors have relegated this experiment to the supplement, I think it is important. This to me is a tell-tale signature of the in-plane magnetic field creating a Fraunhofer-like current distribution across the cross-section of the Josephson junction. Did the authors try reversing the direction of B_y ? Does that change the sign of this asymmetry? Similar effects have been observed in InAs-Al Josephson junctions: (Ref. 21, PRB 95, 035307 (2017), Rasmussen et al. PRB B 93.15, 155406(2016)). The authors mention flux-focussing discussed in Ref. 21 but do not discuss the in-plane Fraunhofer effect in much detail. I think it is important to discuss this effect. More importantly, the authors should comment on whether this effect (similar in spirit to the orbital effect I mentioned above) can influence their diode-effect signatures.

- As mentioned also in the answer to point 2, the effect of flux is minimized in our analysis as we consider the diode effect only where the net out-of-plane flux is zero [Fig.4(c)] and the in-plane fields used in our experiments are much smaller than that required to produce Fraunhofer interference oscillations. The in-plane flux is essentially within a small fraction of the first lobe and the effect of an in-plane flux are quite different from that of δ .
- The change in the direction of the in-plane magnetic field does indeed reverse the observed asymmetry [Refer Fig. S9]. However, we think this does not significantly influence the diode effect, as can be seen in Fig.3(d) and (l). Though a strong asymmetry in the Fraunhofer pattern is observed in Fig.3(d), the diode effect remains symmetric on both sides, indicating that that this asymmetry has little effect in the creation of a JDE.
- This is also clearly visible in Fig.3(a) of Ref. 21, PRB 95, 035307 (2017), in which there is no noticeably strong difference in the positive and negative critical currents despite the strong asymmetry of the interference pattern in positive and negative B_z even for large values of B_y .

7. By performing a Fraunhofer analysis of the critical current asymmetry with increasing B_y , the authors observe an interesting structure of ΔI_c with out-of-plane flux. This structure changes with increasing B_y , which is modeled as a phase difference between the first and second harmonics of the current phase relation. Although the theory and experiment seem to qualitatively agree, quantitatively I don't see a strong

match. Particularly, the $\Phi_0/2$ disappearance of ΔI_c , although clearly observed in theory is very hard to see in the experimental data. Given that the authors claim a "Second order Φ_0 Josephson effect", I think more convincing analysis or data is needed. In the very least, the authors should clearly point out the $\Phi_0/2$ features that the readers are meant to observe in the data.

We would like to note there is a strong agreement between the simulations of $I_{c,2}$ versus S with that observed by experiment in the absence of any flux [Fig.4(c,d)], where flux-related discrepancies do not exist. The discrepancies between the simulation and experiment in some features that arise due to flux induced effects such as lifted nodes and flux focusing, which can't be avoided in such a lateral junction geometry have also been discussed in detail to provide a better understanding of the data to the reader.

The presence of strong second harmonic supercurrents is also confirmed by the quadratic scaling of the extracted second harmonic supercurrents with the critical current of the junction [Fig. 5(f)]. We believe that there is ample evidence provided for substantiating the existence of a 'tunable second order v_0 – Josephson effect' in these junctions.

We have also modified Fig. 4e to better represent the features in periodicity of $I_{c,2}$ and $I_{c,1}$ clearly to the reader.

Fig.4 (e): The evolution of $I_{c,2}$ (in blue) and $I_{c,1}$ (in red) with Φ with $B_y = 20$ mT. Inset shows a close up of oscillations in $I_{c,2}$ has nodes appearing roughly at half magnetic flux quantum ($\Phi_0/2$) frequency (denoted by blue arrows) and has almost double the frequency compared to the nodes in the critical current (denoted by red arrows) that happens roughly at every magnetic flux quantum (Φ_0). It is clearly noticeable that there are twice as many blue arrows compared to red arrows. Black arrows indicate the position of the antinodes

in I_c . The position of the nodes are slightly altered from half magnetic flux quantum due to the presence of lifted nodes and varying flux focusing factor with increasing magnetic flux.

8. In fact, this discrepancy becomes clearer by comparing Figs. 4(f) and 4(g) where I_{c+} is compared with the first Fraunhofer lobe for theory and experiment. Although the "dip" at zero flux is captured very well by theory, two peaks of I_{c+} occur at roughly $\Phi = \Phi_0/2$ in theory, but $\Phi = \Phi_0$ in the experiment. Can the authors comment?

We believe the origin of this discrepancy between Fig. 4(f) and Fig. 4(g) is the variability of the flux focusing factor. The flux focusing effect strongly depends on the junction geometry and varies with the magnetic field. As argued in PRB 95, 035307 (2017), the flux focusing factor (Γ) depends on the magnetic flux and is more pronounced for small B_z (where the junction repels the magnetic field from its interior) and diminishes as B_z increases.

For magnetic flux values $\Phi > 2\Phi_0$, a Γ value of 1.9 accurately describes the nodes in the Fraunhofer pattern shown in Fig. S8(b). Consequently, for $\Phi < 2\Phi_0$, we anticipate a stronger flux focusing effect, as also shown in Fig. S3 of PRB 95, 035307 (2017). This observation resolves the discrepancy noted by the referee, as setting $\Gamma \approx 3$ in Fig. 4(f) would provide a better match with the simulations in Fig. 4(g). We have noted this also in the manuscript.

9. The authors mention large ratios of g/vF in the Discussion section but do not provide any quantitative estimate. Can the authors estimate this factor (even within an order of magnitude) using their experimental values of ΔI_c , for instance?

The Cooper pair momentum due to the Zeeman effect is given by: $2q = \frac{2g\mu B_z}{\hbar v_F}$ (Ref.4,24,25)

The diode effect is maximized when $\delta = \frac{\pi}{2}$, that is $2q \cdot d = \frac{\pi}{2}$ which gives $2q \approx 2.345 \times 10^6 \text{ m}^{-1}$ when $B_z = 24 \text{ mT}$ and hence the value of d to be in the range of 10^{-3} m .

10. I think a schematic showing the spin-momentum locking structure on the Fermi surface of the material would be helpful to the reader, especially since the authors seem to suggest that the Fermi surface does not display usual Rashba-type spin-momentum locking. I also wonder whether the more complicated electronic structure plays any role in the "protection from backscattering" that the authors indicate in the schematic of Fig. 5(g). For example, if an electronic channel with opposite spin-momentum helicity is also available at the Fermi surface (which seems to be the case?), the backscattering protection mechanism should break down.

According to reports in the literature and as also mentioned in the paper, [*Science Bulletin* **64**, 1044-1048 (2019)] PtTe₂ has a Dirac-cone like feature close to the Fermi level, which we think contributes to the observed higher order Andreev reflections in the junction despite the ex-situ fabrication of the superconducting interfaces. There is no counterpart with opposite helicity present at the Fermi level in this work. Though the schematic presented in the paper already well represents this feature, we have modified the text to make this point clearer.

"In the case of PtTe₂, the decreased transparency is due to the contribution of diffusive channels in parallel to transparent ballistic channels. The presence of relatively high transparency despite the ex-situ fabrication of junction interface can be attributed to the significant contribution of the states with helical spin-momentum locking to the transport of supercurrents. We conjecture that the states in PtTe₂ with a spin-momentum locked Dirac-like dispersion¹ can suppress normal reflections due to the reduced availability of spin conserving states. This leads to coherent Andreev processes over long distances leading

to strong second harmonic supercurrents as has been reported previously in other topological systems²⁻⁴ and also depicted in Fig. 5g. For instance, an electron in PtTe₂ moving to the right with an up spin, incident on the surface of the superconductor can be reflected as a hole moving to the left into the same band with opposite spin, which corresponds to the Andreev reflection process. Whereas for the normal reflection process which conserves spin upon reflection, the availability of spin states for the reflected electron is strongly dependent on the incident angle, in a system with spin-momentum locked Dirac-like dispersion as shown previously² and can be highly suppressed for incidences close to the normal. Similar arguments have also been presented in another recent work on highly transparent Dirac semimetal MoTe₂ junctions⁵.”

- 1 Deng, K. *et al.* Crossover from 2D metal to 3D Dirac semimetal in metallic PtTe₂ films with local Rashba effect. *Science Bulletin* **64**, 1044-1048 (2019). <https://doi.org/10.1016/j.scib.2019.05.023>
- 2 Li, C. *et al.* 4 π -periodic Andreev bound states in a Dirac semimetal. *Nat. Mater.* **17**, 875-880 (2018). <https://doi.org/10.1038/s41563-018-0158-6>
- 3 Ando, T., Nakanishi, T. & Saito, R. Berry's Phase and Absence of Back Scattering in Carbon Nanotubes. *J. Phys. Soc. Jpn.* **67**, 2857-2862 (1998). <https://doi.org/10.1143/JPSJ.67.2857>
- 4 Veldhorst, M. *et al.* Josephson supercurrent through a topological insulator surface state. *Nature Mater.* **11**, 417-421 (2012). <https://doi.org/10.1038/nmat3255>
- 5 Zhu, Z. *et al.* Phase tuning of multiple Andreev reflections of Dirac fermions and the Josephson supercurrent in Al-MoTe₂-Al junctions. *Proc. Natl. Acad. Sci. U.S.A.* **119**, e2204468119 (2022). <https://doi.org/10.1073/pnas.2204468119>

Reply to Reviewers' comments

Reviewers' comments:

Reviewer #1 (Remarks to the Author):

The authors have answered the Referees' questions in an appropriate fashion. I confirm my positive judgement and I recommend publication in "Communications physics".

We thank the reviewer for their recommendation and their positive feedback, which has helped strengthen our case and convey the message clearly to the reader.

Reviewer #2 (Remarks to the Author):

I thank the authors for their thoughtful and detailed responses. The authors have addressed more or less all my concerns. However, one issue still bothers me. The g/v_F ratio estimated by the authors is $\sim 10^{-3}$ m/s. Given that $v_F \sim 3.3 \times 10^5$ m/s in this material, we get $g \sim 330$! This is an astoundingly large g -factor, much larger than the $g \sim 50$ that the authors cite in Ref. 58. The authors should mention this clearly in the manuscript and provide a suitable justification for the readers. It is important because this is central to the spin-orbit interpretation of the diode effect. I am happy to recommend publication with this change included.

We thank the reviewer for their questions and positive feedback, which has helped strengthen our case and convey the message clearly to the reader.

An accurate theoretical estimate of the g -factor requires first-principles calculations of the band structure, which would account for both orbital and multiband contributions to the g -factor. For the purpose of this discussion, assuming the Fermi velocity of the bands contributing to superconductivity is $v_F \sim 3.3 \times 10^5$ m/s from literature, we can estimate the g -factor of the electrons in PtTe₂ from the Cooper pair momentum. The Cooper pair momentum is $2q \sim 2.345 \times 10^6$ m⁻¹ when $B_y = 24$ mT. From this, the

value of $(v_F \lambda_F)$ can be estimated to be around 6.3×10^{-4} m⁻¹s. This estimate indicates that the g -factor is around 208, which is a rather large number and one of the largest values reported to date. However, it should be noted that this estimate depends on the assumed value of v_F and the accurate determination of $d_{\text{eff}} = d + 2A$ for the junction, since $g = \frac{h v_F \tau}{4 d_{\text{eff}} \mu_B}$.

In our junctions, the thickness of the superconducting leads is comparable to the coherence length of niobium, which means that A in our superconducting wire geometry can be significantly larger than that of the A of thin film niobium reported in literature. In our calculation of $(v_F \lambda_F)$, we have assumed $A = 100$ nm from literature. Therefore, $(v_F \lambda_F) \sim 6.3 \times 10^{-4}$ m⁻¹s represents rather an upper limit estimate, for the assumed v_F . If we were to assume, for instance, A of the superconducting electrodes to be around 500 nm instead, this would result in a g -factor of 88. Hence, there is a very large uncertainty in the estimation of g -factor from this method, which depends on the accurate estimation of A , which in turn depends strongly on various parameters like the exact thickness, geometry and method of deposition of the superconducting electrodes. The estimation of the exact value of A is further complicated by flux focusing effects in the junction.

It is to be noted that g -factor values as high as 40 have been reported previously in the Dirac semimetal Cd_3As_2 [*Nature Mater* **13**, 851–856 (2014)] and g -factors of more than 100 have been reported in short-period thin film $\text{InAsSb}/\text{InSb}$ superlattices through magneto-absorption measurements [*Nat Commun* **13**, 5960 (2022)]. There are also other references of large g -factors observed in topological semimetals that is provided in the main manuscript.

Though we cannot precisely estimate the g -factor in this material from the diode effect, there have been multiple experiments recently that have demonstrated the existence of extremely large spin-orbit coupling effects in PtTe_2 . It has been shown to possess a very large spin Hall angle and spin-orbit torques that are much larger than other topological semimetals like WTe_2 , Bi_2Se_3 comparable to metallic Pt [*Adv. Mater.* 2020, 32, 2000513; *Results in Physics*, 2024, p. 107630] and has also been used in magnetic switching of ferromagnets [*Nat. Mater.* **23**, 768–774 (2024)]. While the presence of large spin-orbit coupling in PtTe_2 is clear, how its band structure correlates with its unusually large g -factor is something that needs detailed investigation in the future.

We have included this discussion on the g -factor estimation as a supplementary note (as given below) with reference in the main manuscript and hope to have provided a satisfactory response to the reviewer.

“An accurate theoretical estimate of the g -factor requires first-principles calculations of the band structure, which would account for both orbital and multiband contributions to the g -factor. For the purpose of this discussion, assuming the Fermi velocity of the bands contributing to superconductivity is $v_F \approx 3.3 \times 10^5 \text{ m s}^{-1}$, we can estimate the g -factor of the electrons in PtTe_2 from the Cooper pair momentum.

$$2 g \mu_B \gamma$$

The Cooper pair momentum due to the Zeeman effect is given by: $2q \hbar v_F$

The diode effect is maximized

when $\delta = \pi/2$, that is $2q \cdot d_{eff} = 2 \frac{\hbar v_F}{\mu_B}$, which in case of device BI occurs when

$B_y = 24 \text{ mT}$. This gives the Cooper pair momentum to be: $2q \approx 2.345 \times 10^6 \text{ m}^{-1}$ when $B_y = 24 \text{ mT}$

From this, the value of (g_{vF}) can be estimated to be around $6.3 \times 10^{-4} \text{ m}^{-1} \text{ s}$. This estimate indicates that the g -factor is rather large, around 208. However, it should be noted that this estimate depends on the assumed value of v_F and the accurate determination of $d_{eff} = d + 2\lambda$ for the junction, since $g = \frac{\hbar v_F \pi}{4 d e f f \mu_B y}$.

In our junctions, the thickness of the superconducting leads is comparable to the coherence length of niobium, which means that λ in our superconducting wires can be significantly larger than that of the penetration depth of thin film niobium reported in literature. In our calculation of (g_{vF}) , we have assumed $\lambda = 100 \text{ nm}$ from literature. Therefore, $(g_{vF}) \approx 6.3 \times 10^{-4} \text{ m}^{-1} \text{ s}$ represents rather an upper limit

estimate, for the assumed v_F . If we were to assume, for instance, λ of the superconducting electrodes to be around 400 nm , this would result in a g -factor of 88. Hence, there is a very large uncertainty in the estimation of g -factor from this method, which depends on the accurate estimation of λ , which in turn depends strongly on various parameters like the exact thickness, geometry and method of deposition of the superconducting electrodes. The estimation of the exact value of λ is further complicated by flux focusing effects in the junction.

Though we cannot precisely estimate the g -factor in this material from the JDE, we would like to note that there have been recent experiments that have demonstrated the existence of extremely large spin-orbit

coupling effects in PtTe₂. It has been shown to possess a very large spin Hall angle and spin-orbit torques that are much larger than other topological semimetals like WTe₂, Bi₂Se₃ comparable to metallic Pt [Adv. Mater. 2020, 32, 2000513; Results in Physics, 2024, p. 107630] and has also been used in magnetic

switching of ferromagnets[*Nat. Mater.* 23, 768–774 (2024)]. While the presence of large spin-orbit coupling in PtTe_2 is clear, how its band structure correlates with its unusually large g -factor is something that needs detailed investigation in the future.”